# ElasTST: Towards Robust Varied-Horizon Forecasting with Elastic Time-Series Transformer

**Jiawen Zhang**[*]
DSA, HKUST(GZ)
Guangzhou, China
jiawe.zh@gmail.com

**Shun Zheng**[†]
Microsoft Research Asia
Beijing, China
shun.zheng@microsoft.com

**Xumeng Wen**
Microsoft Research Asia
Beijing, China
xumengwen@microsoft.com

**Xiaofang Zhou**
CSE, HKUST
Hong Kong SAR, China
zxf@ust.hk

**Jiang Bian**
Microsoft Research Asia
Beijing, China
jiang.bian@microsoft.com

**Jia Li**[†]
DSA, HKUST(GZ)
Guangzhou, China
jialee@ust.hk

## Abstract

Numerous industrial sectors necessitate models capable of providing robust forecasts across various horizons. Despite the recent strides in crafting specific architectures for time-series forecasting and developing pre-trained universal models, a comprehensive examination of their capability in accommodating varied-horizon forecasting during inference is still lacking. This paper bridges this gap through the design and evaluation of the Elastic Time-Series Transformer (ElasTST). The ElasTST model incorporates a non-autoregressive design with placeholders and structured self-attention masks, warranting future outputs that are invariant to adjustments in inference horizons. A tunable version of rotary position embedding is also integrated into ElasTST to capture time-series-specific periods and enhance adaptability to different horizons. Additionally, ElasTST employs a multi-scale patch design, effectively integrating both fine-grained and coarse-grained information. During the training phase, ElasTST uses a horizon reweighting strategy that approximates the effect of random sampling across multiple horizons with a single fixed horizon setting. Through comprehensive experiments and comparisons with state-of-the-art time-series architectures and contemporary foundation models, we demonstrate the efficacy of ElasTST's unique design elements. Our findings position ElasTST as a robust solution for the practical necessity of varied-horizon forecasting. ElasTST is open-sourced at https://github.com/microsoft/ProbTS/tree/elastst.

## 1  Introduction

Time-series forecasting plays a crucial role in diverse industries, where it is essential to provide forecasts over various time horizons, accommodating both short-term and long-term planning requirements. This includes predicting COVID-19 cases and fatalities one and four weeks ahead to allocate public health resources [7], estimating future electricity demand on an hourly, weekly, or monthly basis to optimize power management [16], and projecting both immediate and long-term traffic conditions for efficient road management [2, 27], among others.

---

[*]This work was done during the internship at Microsoft Research Asia.
[†]Corresponding Author.

38th Conference on Neural Information Processing Systems (NeurIPS 2024).

Despite this, a majority of advanced time-series Transformer [30] variants developed in recent years still necessitate per-horizon training and deployment [37, 40, 33, 32, 22, 20, 39]. These models struggle to handle longer inference horizons once trained for a specific horizon, and may yield suboptimal performance when assessed for shorter horizons. These constraints lead to the practical inconvenience of maintaining distinct model checkpoints for different forecasting horizons required by real-world applications.

Even though recent studies on pre-training universal time-series foundation models have made some progress in facilitating varied-horizon forecasting [26, 9, 8, 31], they primarily concentrate on assessing the overall transfer performance from pre-training datasets to zero-shot scenarios. However, they lack an in-depth investigation into the challenges of generating robust forecasts for different horizons. To be specific, TimesFM [9], a decoder-only Transformer, is capable of arbitrary-horizon forecasting, but this approach could potentially lead to substantial error propagation in long-term forecasting scenarios due to autoregressive decoding. DAM [8], though free from this issue thanks to a novel output design composing sinusoidal functions, cannot effectively capture abrupt changes in time-series data, thereby limiting its utility in critical domains such as energy and traffic. Moreover, while MOIRAI [31] employs a full-attention encoder-only Transformer architecture and supports arbitrary-horizon forecasting via a non-autoregressive manner by introducing mask tokens into forecasting horizons, it remains uncertain how well MOIRAI adapts to different horizons. For example, its architecture design does not ensure the *horizon-invariant* property: the model output for a specific future position should be invariant to arbitrary extensions in forecasting horizons beyond that. Besides, its performance could drop significantly for moderate context lengths.

To address this research gap, we introduce a comprehensive study to explore how to construct a time-series Transformer variant that can yield robust forecasts for varied inference horizons once trained. We name the developed model as *Elastic Time-Series Transformer* (ElasTST). ElasTST adopts a non-autoregressive design by incorporating placeholders into forecasting horizons, which is inspired by diffusion Transformers [24] and the success of SORA [3] in video generation. Here we impose structured self-attention masks, only allowing placeholders to attend to observed time-series patches. This design ensures the aforementioned *horizon-invariant* property by blocking the information exchange across placeholders. Additionally, we devise a tunable version of rotary position embedding (RoPE) [28] to capture customized period coefficients for time series and to learn the adaptation to varied forecasting horizons. Furthermore, we introduce a multi-patch design to balance fine-grained patches beneficial to short-term forecasting with coarse-grained patches preferred by long-term forecasting, and use a shared Transformer backbone to handle these multi-scale patches. Alongside core model designs, during the training phase, we deploy a horizon reweighting approach that approximates the effects of random sampling across multiple training horizons using just one fixed horizon, eliminating the need for additional sampling efforts. Collectively, these key customizations facilitate ElasTST to produce consistent and accurate forecasts across various horizons.

Our extensive experiments affirm the effectiveness of ElasTST in varied-horizon forecasting. First, we evaluated ElasTST, trained with a fixed horizon and employing a reweighting scheme, against state-of-the-art models trained for specific inference horizons. The results demonstrate that ElasTST delivers competitive performance without requiring per-horizon tuning. Then, we examined varied-horizon forecasting for these models, and the advantages of ElasTST are much more outstanding, demonstrating remarkable extrapolations to longer horizons while preserving robust results for shorter ones. Moreover, we also compared ElasTST with some pre-trained time-series models, such as TimesFM and MOIRAI, and found that dataset-specific tuning still offers prominent advantages over zero-shot inference in challenging datasets, such as Weather and Electricity, and that ElasTST can provide more robust performance across different forecasting horizons. At last, we conducted comprehensive ablation tests to highlight the significance of each unique design element of ElasTST.

In summary, our contributions comprise:

- Conducting a systematic study on varied-horizon forecasting, a critical requirement across various domains, yet an underexplored area in time-series research.

- Developing a novel Transformer variant, ElasTST, which incorporates structured attention masks for horizon-invariance, tunable RoPE for time-series-specific periods, multi-patch representations to balance fine-grained and coarse-grained information, and a horizon reweighting scheme to effectively simulate varied-horizon training.

- Demonstrating the effectiveness of ElasTST through experiments comparing it with state-of-the-art time-series architectures and some up-to-date foundation models. Our ablation tests further reveal the importance of its key design elements.

## 2   Related Work

**Traditional Neural Architecture Designs for Time-Series Forecasting**   The field of time-series forecasting has witnessed a significant evolution of neural architectures, transitioning from early multi-layer perceptrons [23], convolutional [5], and recurrent networks [6], to a more recent focus on various Transformer variants [37, 40, 33, 32, 22, 20, 39]. However, the challenge of varied-horizon forecasting remains underexplored in these studies, as these models often require specific tuning to optimize performance for each inference horizon. Additionally, many models, including PatchTST [22], iTransformer [20], and MTST [39], utilize horizon-specific projection heads, which inherently complicates the extension of their forecasting horizons.

**Developing Foundation Models for Time-Series Forecasting**   Inspired by the remarkable successes in the creation of foundational models in the language and vision domains [4, 25, 3], the trend of pre-training universal foundation models has emerged in time-series forecasting research. Notable works in this area include Lag-Llama [26], DAM [8], TimesFM [9], and MOIRAI [31]. These studies employ unique designs to address the challenges posed by varied variate numbers and forecasting horizons when adapting to new scenarios. Lag-Llama, DAM, and TimesFM adopted the univariate paradigm to circumvent the difficulties associated with handling different variates. In contrast, MOIRAI has taken a different approach by flattening multi-variate time series into a single sequence to facilitate cross-variate learning. While this method has its merits, it is worth noting that it may introduce efficiency issues when handling a substantial number of variates and long forecasting horizons. As a result, this paper also adopts the univariate setup to maintain efficiency. When it comes to varied forecasting horizons, Lag-Llama and TimesFM both utilized the decoder-only Transformer and relied on autoregressive decoding to manage arbitrarily long horizons. DAM introduced a novel output scheme that comprises numerous sinusoidal basis functions, enabling it to project into arbitrary future time points. MOIRAI, on the other hand, used a composite input scheme, combining observed time-series patches with variable placeholders that indicate forecasting horizons, and built a full-attention encoder-only Transformer on top of this. Interestingly, this non-autoregressive generation paradigm originates from diffusion transformers used in video generation [24, 3]. In this paper, we also embrace this paradigm for generating variable-length time-series. Unlike MOIRAI, which has made considerable strides in time-series pre-training using a moderately designed Transformer variant, our focus lies in systematically examining critical architectural enhancements to improve robustness in time-series forecasting across various horizons. We believe that constructing a more robust, resilient, and universal architecture will pave the way for more powerful foundational time-series models to be pre-trained in the future.

**Position Encoding in Time-Series Transformers**   Position encoding plays a pivotal role in Transformers as both self-attention and feed-forward modules lack inherent position awareness. The majority of existing time-series Transformer variants have roughly adopted absolute position encoding [30] with minor modifications across different studies. For instance, Informer [40] and Pyraformer [19] have combined fixed absolute position embeddings with timestamp embeddings such as day, week, hour, minute, etc. Meanwhile, Autoformer [33] and Fedformer [41] have omitted absolute position embeddings and relied solely on timestamp embeddings. Other models like Log-Trans [18] and PatchTST [22] have explored learnable position embeddings. However, the challenge with absolute position embedding is its inability to extrapolate into unseen horizons, posing a significant challenge for varied-horizon forecasting. To address this issue, MOIRAI has utilized a relative position embedding technique, RoPE [28], which has been broadly adopted in the language domain to handle variable-length sequences [29]. In our work, we also adopt RoPE to introduce relative position information into self-attention operations. What we uniquely reveal is that the direct application of the RoPE configuration from the language domain to time-series forecasting is not ideal. The reason being that the predefined coefficients do not align well with the typical periodic patterns observed in time-series data. As a solution, we suggest redefining the period range encompassed by the initial RoPE coefficients and making data-driven adjustments to these coefficients.

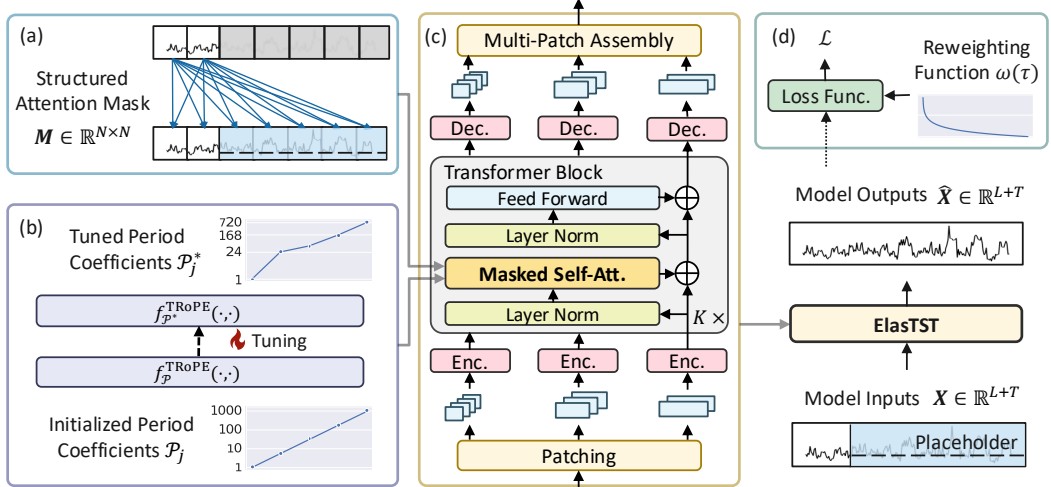

Figure 1: Overview of the ElasTST Architecture. ElasTST employs (a) structured attention masks for placeholders to ensure consistent outputs across varied forecasting horizons. It incorporates (b) tunable RoPE customized to time series periodicities, enhancing its robustness. The architecture also integrates a (c) multi-scale patch assembly that merges fine-grained and coarse-grained details for improved forecasting accuracy. Furthermore, we implement (d) training horizon reweighting scheme during the training phase, which effectively simulates random sampling of forecasting horizons, reducing the need for additional sampling efforts.

**Input Patches in Time-Series Transformers**   PatchTST [22] spearheaded the concept of segmenting time-series data into patches instead of feeding raw time-series values directly into Transformer models. This straightforward yet effective approach has been widely adopted in subsequent studies, including MTST [39], TSMixer [10], HDMixer [17], and MOIRAI. It noteworthy that MOIRAI has been trained with a diverse range of time-series patches with varying patch sizes. When adapting it to a new dataset, practitioners need to search through a range of patch sizes and rely on validation performance to select a single patch size. In our work, however, we have demonstrated that segmenting time series into multiple patch sizes to create multi-scale patch representations is more advantageous. This approach further aids in stabilizing accurate forecasting across various horizons.

## 3   Elastic Time-Series Transformers

In Figure 1, we present an overview of ElasTST. Different from other encoder-only Transformer architectures, ElasTST equipped three core designs to facilitate varied-horizon forecasting: structured self-attention masks for placeholders, tunable rotary position embedding (TRoPE) with custimized period coefficients, and a multi-scale patch representation learning. Additionally, we utilize a horizon reweighting scheme to achieve the effects of varied-horizon training.

**Notations**   We define a univariate time series as $\boldsymbol{x}_{1:T} = \{x_t\}_{t=1}^T$, with $x_t \in \mathbb{R}$ indicating the value at time index $t$. The learning objective of a varied-horizon forecasting can be formulated as: $\max_\phi \mathbb{E}_{\boldsymbol{x} \sim p(\mathcal{D}), (t,L,T) \sim p(\mathcal{T})} \log p_\phi(\boldsymbol{x}_{t+1:t+T} | \boldsymbol{x}_{t-L+1:t})$, where $p(\mathcal{D})$ is the data distribution from which time series samples are drawn, and $p(\mathcal{T})$ is the task distribution, from which the timestamp $t$, look-back window $L$, and the prediction horizon $T$ are sampled.

**Model Inputs**   To accommodate varied forecast horizons, our model combines the historical context series $\boldsymbol{x}_{t-L+1:t}$ with placeholders $\boldsymbol{0} \in \mathbb{R}^T$ through concatenation, forming the input $X = \text{Concat}(\boldsymbol{x}_{t-L+1:t}, \boldsymbol{0})$. This approach allows for flexible adjustment of the input and output dimensions to suit different forecasting scenarios. We further segment $X$ into non-overlapping patches $X^p \in \mathbb{R}^{N \times P}$, where $P$ is the patch length and $N = \frac{(L+T)}{P}$ represents the number of patches. Each input patch is then transformed into latent space by the encoder $\boldsymbol{H} = \text{Enc}(X^p), \boldsymbol{H} \in \mathbb{R}^{N \times D}$.

**Masked Self-Attention**   A robust varied-horizon forecasting method should deliver consistent outputs across different forecasting horizons while maintaining high accuracy on unseen horizons. Existing time series Transformers, however, typically directly adapt techniques from video generation and natural language processing without considering the unique characteristics of time series. To address this deficiency, ElasTST modifies a standard Transformer Encoder with two crucial enhancements: structured attention masks and a tunable RoPE to encode relative position information effectively. We formulate the attention scores within a masked self-attention as

$$a_{m,n} = \langle f^{\text{TRoPE}}(\boldsymbol{h}_m \boldsymbol{W}^q, m), f^{\text{TRoPE}}(\boldsymbol{h}_n \boldsymbol{W}^k, n) \rangle \cdot M_{m,n}, \tag{1}$$

where $\boldsymbol{W}^q, \boldsymbol{W}^k \in \mathbb{R}^{D \times d}$ denote the linear mappings for the query and key, respectively. A tunable RoPE $f^{\text{TRoPE}}$ dynamically adjusts the relative position encoding manner to best suit each dataset, with further details provided in the following subsection. The structured attention mask $M_{\cdot,n}$ is set to 0 for patches $X_n^p$ consisting solely of placeholders and 1 otherwise, ensuring that tokens attend only to context-carrying patches. This structured masking, in conjunction with the relative position encoding, prevents the influence of placeholders on prediction outcomes, thus ensuring consistent outputs across varied forecasting horizons.

**Tunable Rotary Position Embedding**   Position embedding is crucial for the attention mechanism to maintain accuracy over unseen horizons. To overcome the limitations of absolute position embedding in extrapolation scenarios, RoPE has been widely adopted in the NLP domain for handling variable-length sequences. It rotates a vector $\boldsymbol{x} \in \mathbb{R}^d$ onto an embedding curve on a sphere in $\mathbb{C}^{d/2}$, with the rotation parameterized by a base frequency $b$. The function is defined as $f^{\text{RoPE}}(\boldsymbol{x}, t)_j = (x_{2j-1} + ix_{2j})e^{ib^{-2(j-1)/d}t}$, where $j \in [1, 2, ..., d/2]$ [34]. Typically in NLP, the base frequency $b$ is set to a constant, such as 10,000. However, due to the unique characteristics of time series data, specific adaptations of RoPE are necessary. In this paper, we propose to use the period coefficients $\mathscr{P}_j = \frac{2\pi}{b^{-2(j-1)/d}}$ for parameterization:

$$f^{\text{TRoPE}}(\boldsymbol{x}, t)_j = (x_{2j-1} + ix_{2j})e^{i\frac{2\pi}{\mathscr{P}_j}t} \tag{2}$$

$$\mathscr{P}_j = \mathscr{P}_{\min}e^{2\alpha(j-1)}, \quad \alpha = \frac{1}{d-2}\ln\left(\frac{\mathscr{P}_{\max}}{\mathscr{P}_{\min}}\right) \tag{3}$$

where $\mathscr{P}_{\min}$ and $\mathscr{P}_{\max}$ represent the predefined minimum and maximum period coefficients, respectively. This formula maintains an exponential distribution but adjusts the range to better align with the periodic characteristics of time series data. By setting $\mathscr{P}_{\min} = 2\pi$ and $\mathscr{P}_{\max} = 2\pi b^{1-\frac{2}{d}}$, this approach mirrors the original RoPE setup.

In addition to adjusting the period range, the distinct and varying periodicities inherent in time series data necessitate more flexible period coefficients. Therefore, in ElasTST, we consider period coefficients $\mathscr{P}$ as tunable parameters, optimizing it along with varied datasets and forecasting horizons. This adaptive approach allows for more precise and effective forecasting across diverse conditions. We provide a detailed exploration of this design in Section D.4, and illustrate the optimized period coefficients for each dataset in Appendix E.2.

**Multi-Scale Patch Assembly**   To ensure robust performance across various forecasting horizons, integrating both fine-grained and coarse-grained features from time series data is essential. Different from earlier multi-patch models that utilize separate processing branches for each patch size [39], ElasTST features a multi-scale patch design within a shared Transformer backbone, capable of both parallel and sequential processing. We chose sequential processing for our implementation, keeping the memory consumption comparable to baselines such as PatchTST. The implications of this design on memory usage are further discussed in Appendix F. Specifically, we define each patch size as $\boldsymbol{p} = \{p_1, \ldots, p_S\}$, with each size corresponding to a dedicated MLP encoder $f_{p_i}^{\text{Enc}} : \mathbb{R}^{p_i} \mapsto \mathbb{R}^D$ and decoder $f_{p_i}^{\text{Dec}} : \mathbb{R}^D \mapsto \mathbb{R}^{p_i}$. The outputs from each size are flattened and then averaged to produce the final forecast $\hat{X}$. During training, losses under individual patch size are calculated and averaged with the assembled forecast losses, to enhance accuracy and consistency across different scales. Further details on the effectiveness of this design is provided in Section 4.2.

**Training Horizon Reweighting**   To effectively manage varied forecasting horizons, training models across multiple horizon lengths, rather than using fixed ones, is a practical approach [31]. In

this study, we propose to use reweighting scheme for loss computation that simulates this process, without the need for additional sampling efforts. Formally, in the conventional implementation, at each training step $s$, a forecasting horizon $T_s$ is randomly selected from the range $[1, T_{\max}]$.[1] Then the loss $\mathcal{L}_s$ at step $s$ is computed as:

$$\mathcal{L}_s = \sum_{\tau=1}^{T_s} \omega(\tau)(x_{t+\tau} - \hat{x}_{t+\tau})^2, \quad \omega(\tau) = \frac{1}{T_s}. \tag{4}$$

Theoretically, the expectation of this random sampling process can be represented as a weighted loss over a fixed horizon $T_{\max}$. To be specific, the expected value of $\omega(\tau)$ is calculated as: $\mathbb{E}[\omega(\tau)] = \frac{1}{T_{\max}} \sum_{T=1}^{T_{\max}} \frac{1}{T}$. We further approximate the reweighting function by harmonic series as:

$$\omega(\tau) \approx \frac{1}{T_{\max}}(\ln(T_{\max}) - \ln(\tau)). \tag{5}$$

By employing this weighted loss $\omega(\tau)$ during training, we replicate the effect achieved by randomly sampling horizons at an infinite number of training steps. In addition, the function $\omega(\tau)$ can be adapted to follow any desired distribution family and can be made differentiable.

## 4 Experiments

To validate the effectiveness of ElasTST, we systematically assess its performance across various forecasting scenarios, benchmarking it against established models. The results, detailed in Section 4.1, showcase ElasTSTs adaptability to diverse forecasting horizons. Subsequently, we perform an extensive ablation study in Section 4.2 to examine the impact of its key designs.[2]

**Datasets**   Our experiments leverage 8 well-recognized datasets, including 4 from the ETT series (ETTh1, ETTh2, ETTm1, ETTm2), and others include Electricity, Exchange, Traffic, and Weather. These datasets cover a wide array of real-world scenarios and are commonly used as benchmarks in the field. Detailed descriptions of each dataset are provided in Appendix C.1. Following the setup described in [33], all models use a standard lookback window of 96, except TimesFM [9] and MOIRAI [31], which utilize extended lookback windows of 512 and 5000, respectively.

**Baselines**   For our comparative analysis, we select 6 representative forecasting models as baselines: (1) Advanced but non-elastic forecasting models, such as iTransformer [20], PatchTST [22], and DLinear [36]; (2) Autoformer [33], which supports varied-horizon forecasting but requires horizon-specific tuning; (3) the cutting-edge time series foundation model like TimesFM [9] and MOIRAI [31], which are pre-trained for general-purpose forecasting across varied horizons. Our analysis primarily assesses the varied-horizon forecasting capabilities, considering their pre-training on subsets of the datasets used.

**Implementation**   ElasTST is implemented using PyTorch Lightning [12], with a training regimen of 100 batches per epoch, a batch size of 32, and a total duration of 50 epochs. We use the Adam optimizer with a learning rate of 0.001, and experiments are conducted on NVIDIA Tesla V100 GPUs with CUDA 12.1. To ensure fairness, we conducted an extensive grid search for critical hyperparameters across all models in this study. The range and specifics of these hyperparameters are documented in Appendix C.2. For parameters not mentioned in the table, we adhered to the best practice settings proposed in their respective original papers. For evaluation, we use Normalized Mean Absolute Error (NMAE) and Normalized Root Mean Squared Error (NRMSE) as they are scale-insensitive and widely accepted in recent studies [23]. More details are in Appendix C.3.

### 4.1 Main Results

**Comparing ElasTST with Horizon Reweighting to Neural Architectures Tuned for Specific Inference Horizons**   Experimental results demonstrate that ElasTST consistently delivers exceptional performance across all horizons without the need for per-horizon tuning. As evidenced in Table 1, ElasTST outperformed SOTA models on diverse datasets including ETTm1, ETTh1, ETTh2,

---

[1]The look-back window $L$ is fixed.

[2]Unless stated otherwise, horizon reweighting scheme is deactivated in ablation study.

Traffic, Weather, and Exchange, despite these models undergoing specific horizon-based training and tuning. This clearly demonstrates ElasTST's inherent robustness and its remarkable capacity to generalize effectively across varied forecasting scenarios.

Table 1: Results (mean$_{\mathrm{std}}$) on long-term forecasting scenarios with the best in **bold** and the second underlined. Each result contains three independent runs with different seeds. During the training phase, ElasTST utilizes a loss reweighting strategy where a single trained model is applied across all inference horizons, where the $H_{\max}$ is set to 720. Other baseline models undergo horizon-specific training and tuning. Additional baseline results are detailed in Appendix D.1.

| | pred len | ElasTST NMAE | ElasTST NRMSE | iTransformer NMAE | iTransformer NRMSE | PatchTST NMAE | PatchTST NRMSE | DLinear NMAE | DLinear NRMSE | Autoformer NMAE | Autoformer NRMSE |
|---|---|---|---|---|---|---|---|---|---|---|---|
| ETTm1 | 96 | $0.273_{000}$ | $\mathbf{0.488}_{000}$ | $\mathbf{0.271}_{000}$ | $0.568_{000}$ | $0.272_{001}$ | $0.565_{001}$ | $0.282_{002}$ | $0.573_{001}$ | $0.388_{001}$ | $0.711_{003}$ |
| | 192 | $\mathbf{0.289}_{000}$ | $\mathbf{0.520}_{000}$ | $0.301_{000}$ | $0.614_{000}$ | $0.295_{001}$ | $0.602_{005}$ | $0.309_{004}$ | $0.617_{003}$ | $0.442_{001}$ | $0.820_{003}$ |
| | 336 | $\mathbf{0.314}_{000}$ | $\mathbf{0.575}_{000}$ | $0.333_{000}$ | $0.668_{000}$ | $0.323_{001}$ | $0.645_{003}$ | $0.338_{008}$ | $0.654_{007}$ | $0.429_{000}$ | $0.774_{001}$ |
| | 720 | $\mathbf{0.346}_{000}$ | $\mathbf{0.645}_{000}$ | $0.376_{000}$ | $0.741_{000}$ | $0.353_{001}$ | $0.700_{005}$ | $0.387_{006}$ | $0.737_{005}$ | $0.440_{000}$ | $0.793_{000}$ |
| ETTm2 | 96 | $0.150_{000}$ | $0.227_{000}$ | $0.137_{000}$ | $0.227_{000}$ | $\mathbf{0.132}_{001}$ | $\mathbf{0.220}_{002}$ | $0.138_{000}$ | $0.226_{000}$ | $0.158_{000}$ | $0.254_{000}$ |
| | 192 | $0.174_{000}$ | $0.264_{000}$ | $0.161_{000}$ | $0.266_{000}$ | $\mathbf{0.157}_{001}$ | $\mathbf{0.259}_{002}$ | $0.163_{003}$ | $0.264_{001}$ | $0.175_{000}$ | $0.283_{000}$ |
| | 336 | $0.191_{000}$ | $0.289_{000}$ | $0.180_{000}$ | $0.293_{000}$ | $\mathbf{0.176}_{000}$ | $\mathbf{0.286}_{000}$ | $0.188_{001}$ | $0.291_{002}$ | $0.191_{000}$ | $0.307_{000}$ |
| | 720 | $0.211_{000}$ | $\mathbf{0.318}_{000}$ | $0.211_{000}$ | $0.330_{000}$ | $\mathbf{0.205}_{001}$ | $0.324_{002}$ | $0.219_{003}$ | $0.327_{002}$ | $0.217_{000}$ | $0.338_{000}$ |
| ETTh1 | 96 | $0.342_{000}$ | $\mathbf{0.619}_{000}$ | $\mathbf{0.321}_{000}$ | $0.626_{000}$ | $0.328_{003}$ | $0.640_{002}$ | $0.352_{011}$ | $0.668_{012}$ | $0.367_{000}$ | $0.656_{000}$ |
| | 192 | $0.364_{000}$ | $\mathbf{0.661}_{000}$ | $\mathbf{0.359}_{000}$ | $0.690_{000}$ | $0.359_{002}$ | $0.705_{001}$ | $0.393_{001}$ | $0.745_{003}$ | $0.392_{000}$ | $0.706_{000}$ |
| | 336 | $\mathbf{0.371}_{000}$ | $\mathbf{0.666}_{000}$ | $0.388_{000}$ | $0.723_{000}$ | $0.384_{002}$ | $0.740_{004}$ | $0.419_{007}$ | $0.778_{009}$ | $0.398_{000}$ | $0.711_{000}$ |
| | 720 | $\mathbf{0.376}_{000}$ | $\mathbf{0.679}_{000}$ | $0.408_{000}$ | $0.735_{000}$ | $0.397_{002}$ | $0.738_{001}$ | $0.502_{029}$ | $0.860_{049}$ | $0.433_{000}$ | $0.739_{000}$ |
| ETTh2 | 96 | $\mathbf{0.158}_{000}$ | $\mathbf{0.239}_{000}$ | $0.177_{000}$ | $0.279_{000}$ | $0.177_{000}$ | $0.281_{001}$ | $0.211_{027}$ | $0.320_{033}$ | $0.203_{000}$ | $0.317_{000}$ |
| | 192 | $\mathbf{0.170}_{000}$ | $\mathbf{0.259}_{000}$ | $0.203_{000}$ | $0.314_{000}$ | $0.201_{001}$ | $0.314_{001}$ | $0.238_{028}$ | $0.353_{030}$ | $0.226_{000}$ | $0.346_{000}$ |
| | 336 | $\mathbf{0.188}_{000}$ | $\mathbf{0.282}_{000}$ | $0.243_{000}$ | $0.372_{000}$ | $0.240_{001}$ | $0.366_{001}$ | $0.284_{008}$ | $0.407_{013}$ | $0.264_{000}$ | $0.398_{000}$ |
| | 720 | $\mathbf{0.215}_{000}$ | $\mathbf{0.319}_{000}$ | $0.264_{000}$ | $0.386_{000}$ | $0.252_{000}$ | $0.371_{000}$ | $0.307_{000}$ | $0.426_{007}$ | $0.287_{000}$ | $0.416_{000}$ |
| Electricity | 96 | $\mathbf{0.085}_{000}$ | $0.777_{000}$ | $0.098_{000}$ | $\mathbf{0.772}_{000}$ | $0.086_{001}$ | $0.816_{005}$ | $0.090_{001}$ | $0.863_{002}$ | $0.140_{000}$ | $0.977_{016}$ |
| | 192 | $0.093_{000}$ | $0.933_{000}$ | $0.106_{000}$ | $\mathbf{0.916}_{000}$ | $0.092_{001}$ | $0.942_{007}$ | $0.095_{001}$ | $0.974_{001}$ | $0.136_{000}$ | $1.017_{000}$ |
| | 336 | $0.100_{000}$ | $1.063_{000}$ | $0.115_{000}$ | $\mathbf{0.985}_{001}$ | $0.100_{000}$ | $1.035_{003}$ | $0.104_{000}$ | $1.066_{004}$ | $0.147_{000}$ | $1.080_{006}$ |
| | 720 | $0.117_{000}$ | $1.289_{000}$ | $0.133_{000}$ | $\mathbf{1.110}_{001}$ | $0.116_{000}$ | $1.213_{003}$ | $0.122_{001}$ | $1.259_{000}$ | $0.159_{000}$ | $1.283_{005}$ |
| Traffic | 96 | $\mathbf{0.195}_{000}$ | $\mathbf{0.461}_{000}$ | $0.246_{000}$ | $0.511_{000}$ | $0.248_{001}$ | $0.527_{001}$ | $0.356_{009}$ | $0.645_{017}$ | $0.293_{000}$ | $0.560_{000}$ |
| | 192 | $\mathbf{0.193}_{000}$ | $\mathbf{0.459}_{000}$ | $0.259_{000}$ | $0.543_{000}$ | $0.245_{001}$ | $0.528_{001}$ | $0.346_{009}$ | $0.628_{009}$ | $0.318_{000}$ | $0.594_{000}$ |
| | 336 | $\mathbf{0.199}_{000}$ | $\mathbf{0.468}_{000}$ | $0.283_{000}$ | $0.571_{000}$ | $0.257_{002}$ | $0.550_{001}$ | $0.350_{008}$ | $0.631_{008}$ | $0.332_{000}$ | $0.630_{000}$ |
| | 720 | $\mathbf{0.218}_{000}$ | $\mathbf{0.497}_{000}$ | $0.275_{000}$ | $0.563_{000}$ | $0.266_{001}$ | $0.559_{001}$ | $0.365_{009}$ | $0.659_{009}$ | $0.341_{003}$ | $0.611_{002}$ |
| Weather | 96 | $\mathbf{0.086}_{000}$ | $\mathbf{0.287}_{000}$ | $0.089_{000}$ | $0.295_{000}$ | $0.087_{002}$ | $0.294_{002}$ | $0.112_{001}$ | $0.316_{000}$ | $0.239_{004}$ | $0.614_{023}$ |
| | 192 | $0.092_{000}$ | $0.312_{000}$ | $0.093_{000}$ | $\mathbf{0.299}_{000}$ | $0.090_{001}$ | $0.299_{001}$ | $0.122_{001}$ | $0.331_{000}$ | $0.213_{000}$ | $0.533_{002}$ |
| | 336 | $0.091_{000}$ | $0.307_{000}$ | $0.096_{000}$ | $\mathbf{0.297}_{000}$ | $0.092_{002}$ | $0.297_{001}$ | $0.130_{002}$ | $0.340_{002}$ | $0.176_{000}$ | $0.413_{001}$ |
| | 720 | $0.093_{000}$ | $0.308_{000}$ | $0.099_{000}$ | $\mathbf{0.298}_{000}$ | $0.094_{001}$ | $0.298_{003}$ | $0.144_{001}$ | $0.358_{002}$ | $0.170_{001}$ | $0.434_{010}$ |
| Exchange | 96 | $0.026_{000}$ | $0.039_{000}$ | $0.025_{000}$ | $0.039_{000}$ | $\mathbf{0.023}_{000}$ | $\mathbf{0.036}_{000}$ | $0.024_{000}$ | $0.037_{000}$ | $0.032_{000}$ | $0.049_{000}$ |
| | 192 | $\mathbf{0.033}_{000}$ | $\mathbf{0.050}_{000}$ | $0.036_{000}$ | $0.056_{000}$ | $0.034_{000}$ | $0.054_{000}$ | $0.035_{000}$ | $0.055_{000}$ | $0.041_{000}$ | $0.065_{000}$ |
| | 336 | $\mathbf{0.041}_{000}$ | $\mathbf{0.062}_{000}$ | $0.048_{000}$ | $0.072_{000}$ | $0.048_{000}$ | $0.076_{000}$ | $0.048_{001}$ | $0.072_{001}$ | $0.056_{000}$ | $0.091_{000}$ |
| | 720 | $\mathbf{0.059}_{000}$ | $\mathbf{0.089}_{000}$ | $0.076_{000}$ | $0.114_{000}$ | $0.072_{000}$ | $0.106_{001}$ | $0.075_{002}$ | $0.118_{004}$ | $0.112_{002}$ | $0.164_{004}$ |

**Comparing the Robustness of Different Models for Varied Inference Horizons** The results clearly position ElasTST as the most robust option for deploying a single, well-trained model across various inference horizons and application scenarios. As demonstrated in Figure 2, ElasTST consistently maintains strong performance across both seen and unseen horizons, underscoring its ability to navigate beyond trained scopes with consistent accuracy across a wide range of forecasts.

In contrast, other models face significant challenges in varied-horizon forecasting. State-of-the-art models like iTransformer and PatchTST excel within their trained horizons but struggle when extended beyond these limits. Models that require horizon-specific tuning, such as Autoformer, often experience abrupt declines in performance, illustrating that scalability alone is insufficient without tailored optimization. TimesFM, with its autoregressive nature, shows substantial error propagation in unseen datasets like ETT and Exchange, and increased errors in pre-trained datasets such as Weather as the horizon extends. While MOIRAI demonstrates strong zero-shot performance on datasets like ETT and Exchange, we find that on the challenging datasets such as Weather and Electricity, dataset-specific tuning still offers advantages. Furthermore, MOIRAIs performance significantly diminishes with shorter context lengths, as discussed in their paper [31]. In comparison, ElasTST operate effectively with much shorter context lengths.

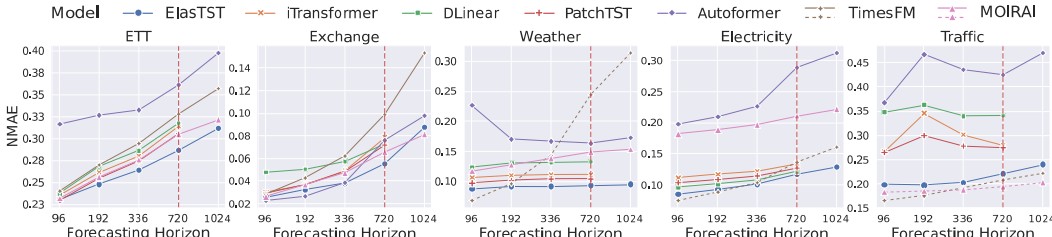

Figure 2: Performance of trained once and inference over varying forecasting horizons. Models except TimesFM and MOIRAI are trained with a forecasting horizon of 720 and tasked with predicting across multiple horizons. A vertical red dashed line distinguishes between their seen horizons (96, 192, 336, 720) and unseen horizon (1024). We use a dashed line to denote the datasets on which the model was pre-trained, e.g., both TimesFM and MOIRAI have leveraged Traffic datasets for their pre-training. The ETT encompasses averaged results from datasets ETTh1, ETTh2, ETTm1, and ETTm2. Models lack inherent elasticity use a truncation strategy for shorter forecasts, and the foundation models use their pre-trained checkpoints and recommended configurations for inference.

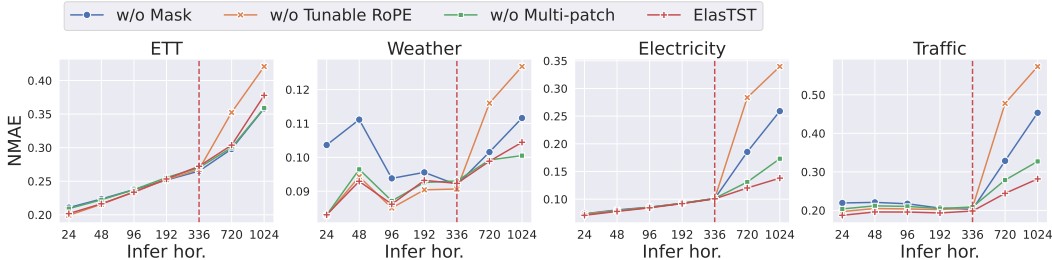

Figure 3: Ablation study for the structured attention masks, tunable RoPE, and multi-patch assembly. A vertical red dashed line indicates the training horizon.

## 4.2 Ablation Study

**Structured Attention Masks** The ablation study confirms that structured attention masks are essential for robust inference across horizons that differ from the training phase. As illustrated in Figure 3, removing structured masks from ElasTST results in significant performance declines, particularly in the Weather dataset. Furthermore, as demonstrated in Figure 7 (see Appendix D.2), the benefits of structured masks are consistent across all forecasting horizons. This underscores the importance of the horizon-invariant property for enhancing the stability of time series forecasting, an aspect often overlooked in current research.

**Tunable Rotary Position Embedding** Experimental results indicate that tunable RoPE significantly improves the models ability to extrapolate. Figure 4a shows that while other positional embedding methods are effective on seen horizons, they falter when applied to horizons extending beyond the training range. Although the original RoPE excels in NLP tasks, it underperforms in time series forecasting. Besides, data-driven adjustments of these coefficients enable far more robust extrapolation. Dynamically tuning RoPE parameters according to the periodic patterns of the dataset proves highly beneficial, especially when inferring over unseen horizons.

Furthermore, a range from 1 to 1000 for the period coefficients $\mathscr{P}$ is more suitable for time series forecasting. As demonstrated in Figure 4b, using the commonly-used NLP settings with $\mathscr{P}_{\min} = 1$ and $\mathscr{P}_{\max} = 10000$ does not fully exploit the potential of RoPE in time series forecasting. Setting $\mathscr{P}_{\max}$ to 1000 results in better performance. We hypothesize that this is because, unlike textual data which benefits from attention over longer contexts, the time series data, especially when segmented into patches, benefits from focusing on shorter, more recent intervals. By adjusting the maximum period coefficient to a lower value, the model captures a richer spectrum of mid-to-high frequency patterns, thereby enhancing its effectiveness. Detailed analyses of these findings are available in Appendix D.4. Appendix E includes visualizations demonstrating how different initial ranges impact frequency components, along with illustrations of the tuned period coefficients for each dataset.

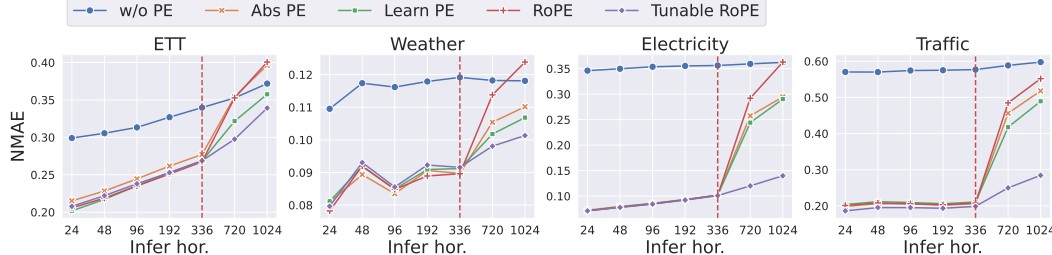

(a) Comparison of various positional embeddings with the proposed Tunable RoPE.

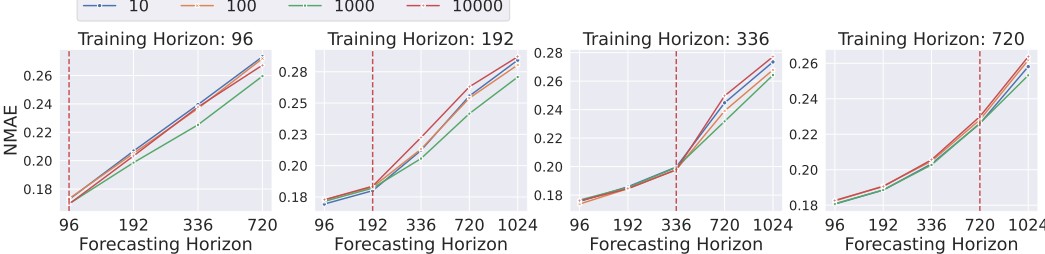

(b) The effect of selecting $\mathscr{P}_{\max}$, with $\mathscr{P}_{\min}$ fixed at 1 and $\mathscr{P}$ is untunable during training.

Figure 4: Ablation study for designs in position embedding. A vertical red dashed line distinguishes between seen horizons and unseen horizons.

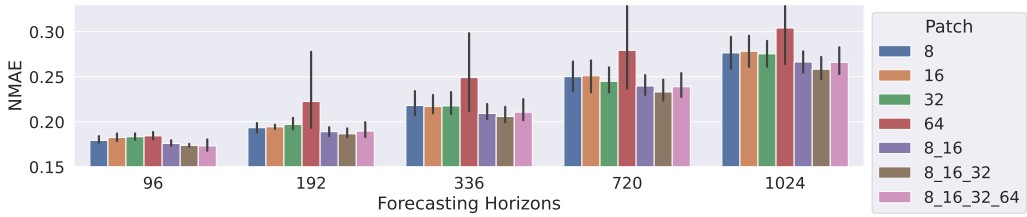

Figure 5: Performance of patch size selections. Results are averaged across all datasets and training horizons of $\{96, 192, 336, 720\}$. '8_16_32' represents a multi-patch configuration of $\boldsymbol{p} = \{8, 16, 32\}$.

**Multi-Patch Design** These experiments demonstrate that multi-patch configurations generally outperform single patch sizes across various forecasting horizons. Figure 5 shows that the configuration $\boldsymbol{p} = \{8, 16, 32\}$ consistently achieves the lowest NMAE values, effectively balancing the capture of short-term dynamics and long-term trends. However, adding larger patches, such as $\boldsymbol{p} = \{8, 16, 32, 64\}$, does not consistently improve performance and can sometimes increase the NMAE. This suggests that more complex configurations may not always provide additional benefits and could even be counterproductive.

Moreover, the patch size selection is particularly critical in the varied-horizon forecasting scenarios. As demonstrated in the Figure 10 (see Appendix D.5), various combinations of training and forecasting horizons exhibit distinct preferences for patch sizes. For instance, when the training forecasting horizon is 720, during the inference stage, longer forecasting horizons prefer larger patch sizes. Conversely, on shorter training horizons, such as 96 and 192, choosing large patch sizes for longer horizons can lead to performance collapse. This difference underscores the complexity and necessity of optimal patch size selection in achieving effective elastic forecasting. Detailed results for four training horizons and further analysis are provided in Appendix D.5.

**The Impact of Training Horizons** Further experiments validate the effectiveness of our proposed training horizon reweighting scheme in enhancing varied-horizon inference. As illustrated in Figure 6, reweighting longer horizons simplifies the training process, yielding better outcomes than

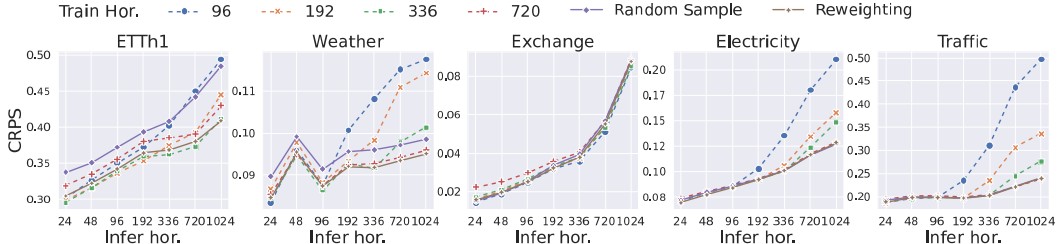

Figure 6: Impact of forecasting horizon selection during the training phase.

selecting a fixed horizon and mitigating the uncertainties associated with random sampling. Crucially, this training approach is model-agnostic and can be applied to different forecasting training scenarios. These results also highlight the advantages of a flexible forecasting architecture, which allows training horizons to be customized to the unique characteristics of each dataset.

We also observe that different datasets have distinct preferences for training horizons. For example, in the Exchange dataset, the longest training horizon led to worse results compared to a shorter horizon of 96, suggesting risks of overfitting or forecast instability with prolonged horizons. Besides, in the ETTh1, employing random sampling for training horizons proved suboptimal. These insights show that tailoring the training horizon selection strategy to the specific dataset can yield improvements. One potential enhancement could involve dynamically optimizing the horizon reweighting scheme alongside model training.

## 5   Conclusion

This study introduces the Elastic Time-Series Transformer (ElasTST), a pioneering model designed to tackle the significant and insufficiently explored challenge of varied-horizon forecasting. ElasTST integrates a non-autoregressive framework with innovative elements such as structured self-attention masks, tunable Rotary Position Embedding (RoPE), and a versatile multi-scale patch system. Additionally, we implement a training horizon reweighting scheme that simulates random sampling of forecasting horizons, thus eliminating the need for extra sampling efforts. Together, these elements enable ElasTST to adapt to a wide range of forecasting horizons, delivering reliable and competitive outcomes even when facing horizons that were not encountered during the training phase.

**Limitations**   While ElasTST demonstrates robust performance across various forecasting tasks, several limitations have been identified that highlight opportunities for future enhancements. First, the current version of ElasTST does not incorporate a pre-training phase, which could significantly improve the models initial grasp of time-series dynamics and boost its efficiency during task-specific fine-tuning. Further exploration is needed to ascertain optimal training methodologies that maximize the architectural benefits of ElasTST. Additionally, while the training horizon reweighting scheme is straightforward and effective in enhancing performance across different inference horizons, it is not the optimal solution for all datasets. Moreover, the evaluation of ElasTST is limited to a select number of datasets, which may not fully represent the broader challenges encountered in more complex or diverse real-world scenarios.

**Future Work**   In response to these limitations, our forthcoming research efforts will concentrate on developing and validating pre-training protocols for ElasTST to elevate its foundational performance and extend its applicability across universal forecasting tasks. We aim to incorporate a reasonable training approach that will fine-tune the models ability to seamlessly manage forecasts of varying lengths, thus bolstering its utility in dynamic real-world environments. Furthermore, by broadening the range of datasets used for model evaluations, we intend to rigorously test ElasTSTs effectiveness across an expanded spectrum of industry-specific challenges. This comprehensive approach will not only solidify ElasTSTs standing as a cutting-edge solution for time-series forecasting but also enhance our understanding of its practical implications and potential in diverse industrial applications.

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

# A  Details of Methods

## A.1  Details of Rotary Position Embedding

Rotary position embedding (RoPE) [28] is a method used to encode the position of tokens in the input sequence for transformer-based models, enhancing the capability to utilize the positional context of tokens. RoPE uniquely incorporates the geometric property of vectors, transforming them into a rotary matrix that interacts with the vector embeddings.

Here, we have adopted the formal definition from the original RoPE paper. In the simplest two-dimensional (2D) case, RoPE considers a dimension $d = 2$, where each position vector is treated in its complex form. The formulation is given by:

$$f_q(x_m, m) = (W_q x_m)e^{im\theta},$$
$$f_k(x_n, n) = (W_k x_n)e^{in\theta},$$
$$g(x_m, x_n, m - n) = \text{Re}[(W_q x_m)(W_k x_n)^* e^{i(m-n)\theta}].$$

This expression shows that the embedding rotates the affine-transformed word embedding vectors by angle multiples relative to their position indices. This rotation is mathematically represented through a multiplication matrix:

$$f_{q,k}(x_m, m) = \begin{bmatrix} \cos m\theta & -\sin m\theta \\ \sin m\theta & \cos m\theta \end{bmatrix} \begin{bmatrix} W_{q,k}^{(11)} & W_{q,k}^{(12)} \\ W_{q,k}^{(21)} & W_{q,k}^{(22)} \end{bmatrix} \begin{bmatrix} x_m^{(1)} \\ x_m^{(2)} \end{bmatrix}.$$

For a generalized form in any dimension $d$, RoPE divides the space into $d/2$ sub-spaces and combines them to utilize the linearity of the inner product. The generalized rotary matrix $R_{\Theta,m}^d$ is defined as:

$$R_{\Theta,m}^d = \begin{bmatrix} \cos m\theta_1 & -\sin m\theta_1 & 0 & 0 & \dots & 0 & 0 \\ \sin m\theta_1 & \cos m\theta_1 & 0 & 0 & \dots & 0 & 0 \\ 0 & 0 & \cos m\theta_2 & -\sin m\theta_2 & \dots & 0 & 0 \\ 0 & 0 & \sin m\theta_2 & \cos m\theta_2 & \dots & 0 & 0 \\ \vdots & \vdots & \vdots & \vdots & \ddots & \vdots & \vdots \\ 0 & 0 & 0 & 0 & \dots & \cos m\theta_{d/2} & -\sin m\theta_{d/2} \\ 0 & 0 & 0 & 0 & \dots & \sin m\theta_{d/2} & \cos m\theta_{d/2} \end{bmatrix},$$

where $\Theta = \{\theta_i = 10000^{-2(i-1)/d}, i \in [1, 2, ..., d/2]\}$. This formulation ensures that RoPE is computationally efficient and stable due to the orthogonality of $R_{\Theta,m}^d$. The use of sparse matrices further improves computational efficiency, making RoPE a practical approach to incorporate in large-scale transformer models.

# B  Additional Related Work on Foundation Models

We summarize existing time series foundational models in Table 2, excluding the LLM-oriented ones [35]. These models typically use standard architecture designs, position encodings, and patching approaches, primarily aiming to enhance transferability in zero-shot scenarios. However, they generally do not deeply explore the challenges of producing robust forecasts across varied horizons. Our work specifically addresses this gap by improving model design to enhance robustness for varied-horizon forecasting.

# C  Additional Details of Experiment Setting

## C.1  Dataset Details

Our experiments utilize 8 widely recognized datasets, including 4 from the ETT series (ETTh1, ETTh2, ETTm1, ETTm2), as well as the Electricity, Exchange, Traffic, and Weather datasets. These

Table 2: Summary of Time Series Foundation Models.

| Model | Backbone | Dec. Scheme. | Pos. Emb. | Token. |
|---|---|---|---|---|
| TimeGPT-1 [14] | Enc-Dec Transformer | AR | Abs PE | - |
| Lag-Llama [26] | Decoder-only Transformer | AR | RoPE | - |
| Chronos [1] | Enc-Dec Transformer | AR | Simplified relative PE | Quantization |
| Timer [21] | Decoder-only Transformer | AR | Abs PE | Patching |
| TimesFM [9] | Decoder-only Transformer | AR | Abs PE | Patching |
| UniTS [13] | Transformer Encoder | NAR | Learnable PE | Patching |
| DAM [8] | Transformer Encoder | NAR | Abs PE | ToME |
| Tiny Time Mixers [11] | TSMixer | NAR | - | Patching |
| MOIRAI [31] | Transformer Encoder | NAR | RoPE | Patching |
| MOMENT [15] | Transformer Encoder | NAR | Learnable relative PE | Patching |

datasets encompass a broad range of real-world applications and are frequently used as benchmarks in the field[3]. Consistent with common practices in long-term forecasting [33, 36, 22, 20], all models are tested under the forecasting horizons $T \in \{96, 192, 336, 720\}$. Except for TimeFM [9], which uses a lookback window of 512, a standard lookback window of 96 is employed across all other models as [33].

Table 3: Dataset Summary.

| Dataset | #var. | range | freq. | timesteps | Description |
|---|---|---|---|---|---|
| ETTh1/h2 | 7 | $\mathbb{R}^+$ | H | 17,420 | Electricity transformer temperature per hour |
| ETTm1/m2 | 7 | $\mathbb{R}^+$ | 15min | 69,680 | Electricity transformer temperature every 15 min |
| Electricity | 321 | $\mathbb{R}^+$ | H | 26,304 | Electricity consumption (Kwh) |
| Traffic | 862 | (0,1) | H | 17,544 | Road occupancy rates |
| Exchange | 8 | $\mathbb{R}^+$ | Busi. Day | 7,588 | Daily exchange rates of 8 countries |
| Weather | 21 | $\mathbb{R}^+$ | 10min | 52,696 | Local climatological data |

## C.2   Implementation Details

ElasTST is implemented using PyTorch Lightning [12]. Training consists of 100 batches per epoch, capped at 20 epochs, with the NMAE metric used for model checkpointing. We use the Adam optimizer with a learning rate of 0.001, and experiments are conducted on NVIDIA Tesla V100 GPUs with CUDA 12.1. The code for Transformer Block is adapted from [38].

**Baselines**   We select five representative forecasting models as our baselines:

- iTransformer [4] [20]: A transformer-based model that inverts the dimensions of time and variates to effectively capture multivariate correlations, enhancing generalization across different variates.
- PatchTST [5] [22]: A transformer-based model segmenting time series into subseries-level patches and employing channel-independent processing to improve forecasting accuracy.
- DLinear [6] [36]: An MLP-based model that has demonstrated superior performance over more complex transformer-based models in multiple real-life datasets.
- Autoformer [7] [33]: Integrates a novel Auto-Correlation mechanism that exploits series periodicity for enhanced dependency discovery and representation aggregation.
- TimeFM [8] [9]: A foundation model for time series forecasting, pretrained using a decoder-style attention mechanism and input patching on an extensive corpus of real-world and synthetic data.

---

[3]Datasets are available at `https://github.com/thuml/Autoformer` under MIT License.

[4]`https://github.com/thuml/iTransformer`, MIT License.

[5]`https://github.com/yuqinie98/PatchTST`, Apache-2.0 license.

[6]`https://github.com/cure-lab/LTSF-Linear`, Apache-2.0 license.

[7]`https://github.com/thuml/Autoformer`, MIT License.

[8]`https://github.com/google-research/timesfm`, Apache-2.0 license.

- MOIRAI (UNI2TS) [9] [31]: A foundation model for time series forecasting, pretrained using a masked encoder-based Transformer on the extensive Large-scale Open Time Series Archive (LOTSA) with over 27 billion observations.

**Hyper-parameter Tuning**    To ensure fairness, we conducted an extensive grid search for critical hyperparameters across all models in this study. The range and specifics of these hyperparameters are documented in Table 4. For parameters not mentioned in the table, we adhered to the best practice settings proposed in their respective original papers.

Table 4: Hyper-parameters values fixed or range searched in hyper-parameter tuning.

| Models | Hyper-parameter | Value or Range Searched |
|---|---|---|
| ElasTST | learning_rate | 0.001 |
| | multi_patch_size | 8-16-32 |
| | min_period_coeff | 1 |
| | max_period_coeff | 1000 |
| | n_heads | [2, 4, 8, 16] |
| | n_layers | [1,2,4] |
| | hidden_size | [128,256,512] |
| | d_v | [64,128] |
| iTransformer | learning_rate | [0.0001,0.0005,0.001] |
| | hidden_size | [128,256,512,1024] |
| | e_layers | [1,2,3,4] |
| PatchTST | learning_rate | [0.0001, 0.001] |
| | patch_len | 16 |
| | stride | 8 |
| | n_layers | 3 |
| | d_model | [16, 128,256,512] |
| | d_ff | [128,256,512] |
| | n_heads | [4,8,16] |
| | dropout | [0.2, 0.3] |
| DLinear | learning_rate | [0.0001,0.0005,0.001, 0.005, 0.05] |
| | kernel_size | 25 |
| Autoformer | learning_rate | [0.0001, 0.001] |
| | e_layers | [1,2,3] |
| | d_layers | [1,2,3] |
| | factor | [1,3] |

## C.3   Evaluation Metrics

We employ the Normalized Mean Absolute Error (NMAE) and Normalized Root Mean Squared Error (NRMSE) as our evaluation metrics because they provide a relative measure of error that is independent of the data scale. It's important to note that some original papers reported metrics prior to re-scaling forecasts to their original magnitude, which can affect metric calculations. In this study, we have carefully ensured that our reproduced results are consistent with those reported in the original studies and have applied these unified metrics to enable a comprehensive and fair comparison.

**Normalized Mean Absolute Error (NMAE)**    The Normalized Mean Absolute Error (NMAE) is a normalized version of the MAE, which is dimensionless and facilitates the comparability of the error magnitude across different datasets or scales. The mathematical representation of NMAE is given by:

$$\text{NMAE} = \frac{\sum_{k=1}^{K} \sum_{t=1}^{T} |x_t^k - \hat{x}_t^k|}{\sum_{k=1}^{K} \sum_{t=1}^{T} |x_t^k|}.$$

---

[9] https://github.com/SalesforceAIResearch/uni2ts, Apache-2.0 license.

**Normalized Root Mean Squared Error (NRMSE)**  The Normalized Root Mean Squared Error (NRMSE) is a normalized version of the Root Mean Squared Error (RMSE), which quantifies the average squared magnitude of the error between forecasts and observations, normalized by the expectation of the observed values. It can be formally written as:

$$\text{NRMSE} = \frac{\sqrt{\frac{1}{K \times T} \sum_{i=1}^{K} \sum_{t=1}^{L} (x_{i,t} - \hat{x}_{i,t})^2}}{\frac{1}{K \times T} \sum_{i=1}^{K} \sum_{t=1}^{T} |x_{i,t}|}.$$

# D  Additional Experimental Results

## D.1  Comparing ElasTST with More Neural Architectures

Table 5: Results (mean$_{\text{std}}$) on long-term forecasting scenarios with the best in **bold** and the second underlined. Each result containing three independent runs with different seeds. During the training phase, ElasTST utilizes an loss reweighting strategy where a single trained model is applied across all inference horizons, the $H_{\max}$ is set to 720. Other baseline models undergo horizon-specific training and tuning.

| | pred len | ElasTST NMAE | ElasTST NRMSE | iTransformer NMAE | iTransformer NRMSE | PatchTST NMAE | PatchTST NRMSE | DLinear NMAE | DLinear NRMSE | TSMixer NMAE | TSMixer NRMSE | Autoformer NMAE | Autoformer NRMSE | Transformer Enc NMAE | Transformer Enc NRMSE |
|---|---|---|---|---|---|---|---|---|---|---|---|---|---|---|---|
| ETTm1 | 96 | 0.273 | **0.488** | **0.271** | 0.568 | _0.272_ | 0.565 | 0.282 | 0.573 | 0.369 | 0.620 | 0.388 | 0.711 | 0.320 | _0.561_ |
| | 192 | **0.289** | 0.520 | 0.301 | 0.614 | _0.295_ | _0.602_ | 0.309 | 0.617 | 0.393 | 0.673 | 0.442 | 0.820 | 0.348 | 0.632 |
| | 336 | **0.314** | **0.575** | 0.333 | 0.668 | _0.323_ | _0.645_ | 0.338 | 0.654 | 0.426 | 0.727 | 0.429 | 0.774 | 0.373 | 0.679 |
| | 720 | **0.346** | **0.645** | 0.376 | 0.741 | _0.353_ | _0.700_ | 0.387 | 0.737 | 0.464 | 0.788 | 0.440 | 0.793 | 0.397 | 0.712 |
| ETTm2 | 96 | 0.150 | 0.227 | _0.137_ | 0.227 | **0.132** | **0.220** | 0.138 | _0.226_ | 0.560 | 0.668 | 0.158 | 0.254 | 0.156 | 0.233 |
| | 192 | 0.174 | _0.264_ | _0.161_ | 0.266 | **0.157** | **0.259** | 0.163 | _0.264_ | 0.556 | 0.665 | 0.175 | 0.283 | 0.183 | 0.275 |
| | 336 | 0.191 | _0.289_ | _0.180_ | 0.293 | **0.176** | **0.286** | 0.188 | 0.291 | 0.553 | 0.664 | 0.191 | 0.307 | 0.201 | 0.303 |
| | 720 | _0.211_ | 0.318 | _0.211_ | 0.330 | **0.205** | _0.324_ | 0.219 | 0.327 | 0.548 | 0.661 | 0.217 | 0.338 | 0.223 | 0.336 |
| ETTh1 | 96 | 0.342 | **0.619** | **0.321** | _0.626_ | _0.328_ | 0.640 | 0.352 | 0.668 | 0.343 | 0.628 | 0.367 | 0.656 | 0.389 | 0.693 |
| | 192 | _0.364_ | **0.661** | 0.359 | _0.690_ | 0.359 | 0.705 | 0.393 | 0.745 | 0.399 | 0.706 | 0.392 | 0.706 | 0.414 | 0.722 |
| | 336 | **0.371** | **0.666** | 0.388 | 0.723 | _0.384_ | 0.740 | 0.419 | 0.778 | 0.449 | 0.749 | 0.398 | _0.711_ | 0.459 | 0.799 |
| | 720 | **0.376** | **0.679** | 0.408 | _0.735_ | _0.397_ | 0.738 | 0.502 | 0.860 | 0.499 | 0.779 | 0.433 | 0.739 | 0.473 | 0.806 |
| ETTh2 | 96 | **0.158** | **0.239** | _0.177_ | 0.279 | _0.177_ | 0.281 | 0.211 | 0.320 | 0.199 | 0.283 | 0.203 | 0.317 | 0.210 | 0.312 |
| | 192 | **0.170** | **0.259** | 0.203 | _0.314_ | _0.201_ | _0.314_ | 0.238 | 0.353 | 0.228 | 0.322 | 0.226 | 0.346 | 0.228 | 0.339 |
| | 336 | **0.188** | **0.282** | 0.243 | 0.372 | _0.240_ | 0.366 | 0.284 | 0.407 | 0.253 | _0.354_ | 0.264 | 0.398 | 0.244 | 0.361 |
| | 720 | **0.215** | **0.319** | 0.264 | 0.386 | _0.252_ | _0.371_ | 0.307 | 0.426 | 0.288 | 0.390 | 0.287 | 0.416 | 0.262 | 0.391 |
| Electricity | 96 | **0.085** | _0.777_ | 0.098 | **0.772** | _0.086_ | 0.816 | 0.090 | 0.863 | 0.101 | 0.862 | 0.140 | 0.977 | 0.107 | 0.924 |
| | 192 | _0.093_ | _0.933_ | 0.106 | **0.916** | **0.092** | 0.942 | 0.095 | 0.974 | 0.105 | 0.951 | 0.136 | 1.017 | 0.109 | 0.957 |
| | 336 | _0.101_ | 1.063 | 0.115 | **0.985** | **0.100** | 1.035 | 0.104 | 1.066 | 0.109 | _1.022_ | 0.147 | 1.080 | 0.118 | 1.057 |
| | 720 | _0.117_ | 1.289 | 0.133 | **1.110** | **0.116** | 1.213 | 0.122 | 1.259 | 0.120 | 1.194 | 0.159 | 1.283 | 0.123 | _1.118_ |
| Traffic | 96 | **0.195** | **0.461** | _0.246_ | _0.511_ | 0.248 | 0.527 | 0.356 | 0.645 | 0.313 | 0.590 | 0.293 | 0.560 | 0.319 | 0.576 |
| | 192 | **0.193** | **0.459** | 0.259 | 0.543 | _0.245_ | _0.528_ | 0.346 | 0.628 | 0.295 | 0.555 | 0.318 | 0.594 | 0.315 | 0.569 |
| | 336 | **0.199** | **0.468** | 0.283 | 0.571 | _0.257_ | _0.550_ | 0.350 | 0.631 | 0.316 | 0.575 | 0.332 | 0.630 | 0.314 | 0.567 |
| | 720 | **0.218** | **0.497** | 0.275 | 0.563 | _0.266_ | _0.559_ | 0.365 | 0.659 | 0.332 | 0.598 | 0.341 | 0.611 | 0.334 | 0.587 |
| Weather | 96 | **0.086** | **0.287** | 0.089 | 0.295 | _0.087_ | _0.294_ | 0.112 | 0.316 | 0.118 | 0.309 | 0.239 | 0.614 | 0.106 | 0.309 |
| | 192 | _0.092_ | _0.312_ | 0.093 | **0.299** | **0.090** | 0.299 | 0.122 | 0.331 | 0.123 | 0.325 | 0.213 | 0.533 | 0.111 | 0.327 |
| | 336 | **0.091** | _0.307_ | 0.096 | **0.297** | _0.092_ | 0.297 | 0.130 | 0.340 | 0.126 | 0.328 | 0.176 | 0.413 | 0.114 | 0.330 |
| | 720 | **0.093** | _0.308_ | 0.099 | **0.298** | _0.094_ | **0.298** | 0.144 | 0.358 | 0.129 | 0.330 | 0.170 | 0.434 | 0.113 | 0.324 |
| Exchange | 96 | 0.026 | 0.039 | 0.025 | 0.039 | **0.023** | **0.036** | _0.024_ | _0.037_ | 0.040 | 0.055 | 0.032 | 0.049 | 0.030 | 0.045 |
| | 192 | **0.033** | **0.050** | 0.036 | 0.056 | _0.034_ | _0.054_ | 0.035 | 0.055 | 0.052 | 0.072 | 0.041 | 0.065 | 0.037 | 0.055 |
| | 336 | **0.041** | **0.062** | _0.048_ | _0.072_ | _0.048_ | 0.076 | _0.048_ | _0.072_ | 0.067 | 0.092 | 0.056 | 0.091 | 0.049 | 0.074 |
| | 720 | **0.059** | **0.089** | 0.076 | 0.114 | _0.072_ | _0.106_ | 0.075 | 0.118 | 0.091 | 0.128 | 0.112 | 0.164 | 0.085 | 0.120 |

## D.2  Performance Gains Across the Forecasting Horizon

In Figure 7, we compare the performance gains of each model design at different points within the forecasting window. The benefits of structured masks are consistent across the entire horizon, while the advantages of tunable RoPE and multi-patch designs become more prominent when handling unseen horizons. Notably, the tunable RoPE plays a critical role in enhancing the models extrapolation capability.

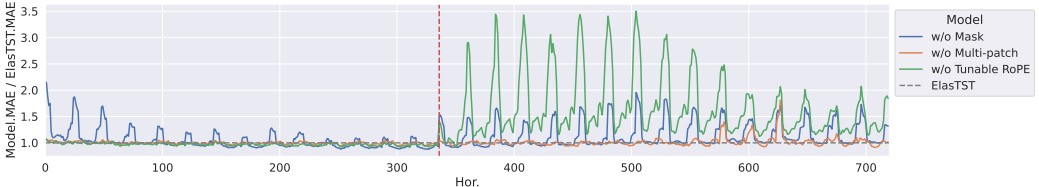

Figure 7: Performance gain of each model design across the forecasting horizon. A relative performance greater than 1 indicates a gain, while values less than 1 indicate a drop. Results are averaged across all datasets, with the vertical red dashed line marking the training horizon.

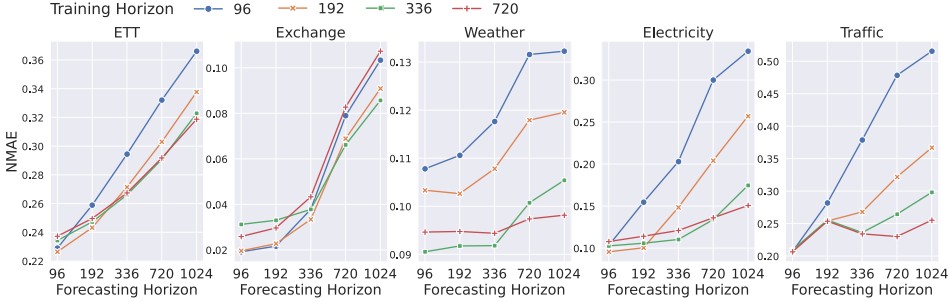

Figure 8: Impact of forecasting horizon selection during the training phase.

## D.3 More Analysis of the Impact of Training Horizon

The scalable architecture of ElasTST allows it to treat the training horizon as a hyperparameter. This adaptability prompts us to evaluate performance across various training and inference horizons (see Figure 8), yielding several interesting insights.

- Model performance deteriorates as the forecasting horizon increases, particularly in models trained on shorter horizons, as seen in the ETT and Electricity datasets. This pattern suggests the importance of training models on extended horizons to capture adequate contextual information.

- Tuning models specifically for a given horizon does not guarantee improved performance on that horizon, as noted in the Weather dataset. This indicates that optimal model settings depend significantly on dataset-specific characteristics, and horizon-specific tuning may not be a reliable strategy.

- The longest training horizons do not always produce the best forecasting results. In the Exchange dataset, for example, the longest horizon yielded poorer results compared to a shorter training horizon of 96. This points to the potential risks of overfitting or forecast instability when training with long-term series only.

These observations underscore the importance of tailoring training horizons to the unique characteristics of each dataset and underscore the benefits of an architecture designed for elastic forecasting. Furthermore, they suggest the potential advantages of implementing a mixed-horizon training strategy, which leverages multiple horizons to produce more resilient forecasts.

## D.4 More Analysis of the Impact of Tunable Rotary Position Embedding

Beyond its scalable architecture, the position embedding plays a crucial role in enhancing the elasticity of ElasTST. We analyze the Tunable RoPE in ElasTST by examining the effects of the initialization of period coefficients, specifically $\mathcal{P}_{\min}$ and $\mathcal{P}_{\max}$, and the benefits of parameter optimization during the training process.

Experimental results, presented in Figure 9, indicate that using settings similar to the commonly-used one in NLP, with $\mathcal{P}_{\min} = 1$ and $\mathcal{P}_{\max} = 10000$, does not fully exploit its potential in time series

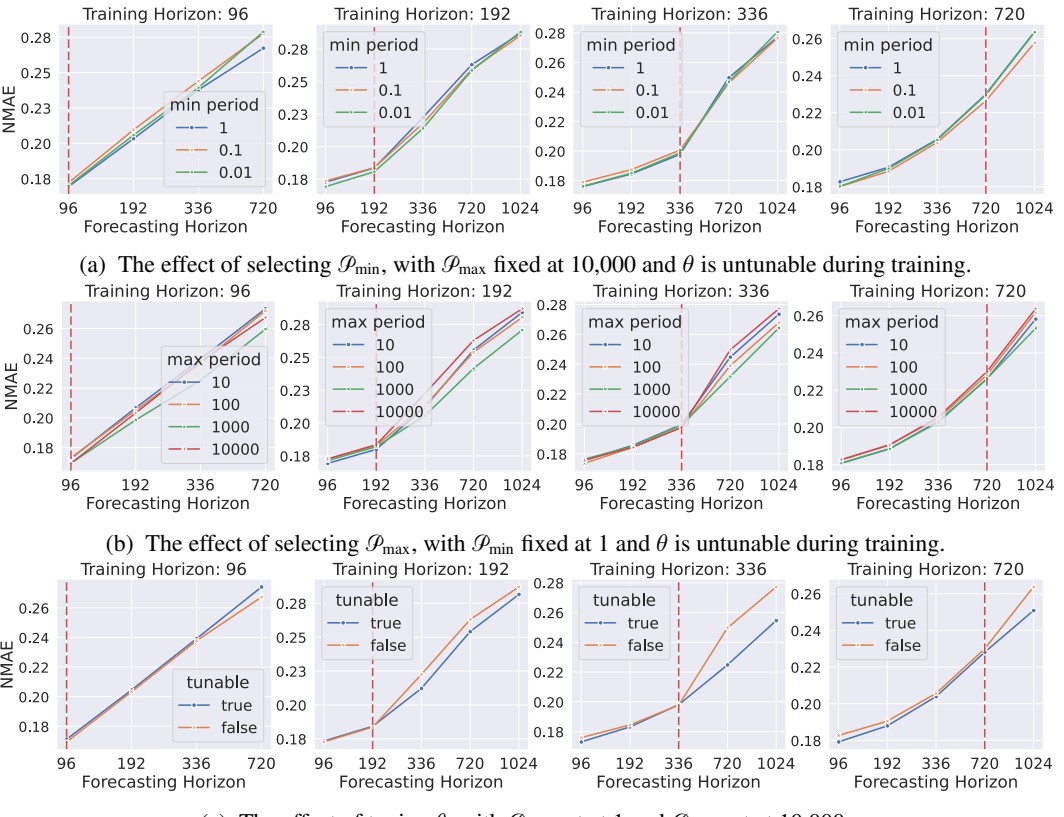

(a) The effect of selecting $\mathscr{P}_{\min}$, with $\mathscr{P}_{\max}$ fixed at 10,000 and $\theta$ is untunable during training.

(b) The effect of selecting $\mathscr{P}_{\max}$, with $\mathscr{P}_{\min}$ fixed at 1 and $\theta$ is untunable during training.

(c) The effect of tuning $\theta$, with $\mathscr{P}_{\min}$ set at 1 and $\mathscr{P}_{\max}$ set at 10,000.

Figure 9: Ablation study for designs in position embedding. Here we analyze the tunable RoPE in ElasTST by examining the effects of the initialization of period coefficients, specifically $\mathscr{P}_{\min}$ and $\mathscr{P}_{\max}$, and the benefits of parameter optimization during the training process. Results are averaged across all datasets. A vertical red dashed line distinguishes between seen horizons and unseen horizons.

forecasting. This discrepancy stems from fundamental differences between the data types: in text, discrete tokens are the smallest units, requiring attention over longer contexts, while time series data, particularly when patched, may benefit from focusing on shorter, more recent tokens.

Furthermore, our findings reveal that tuning parameters in RoPE during training significantly improves forecasting accuracy, particularly over varying and extended horizons. When the training horizon is set to 96, a tunable feature shows minimal impact, suggesting that a short-seen horizon does not facilitate learning effective period coefficients. However, as the training horizon extends, the advantages of a tunable theta become more pronounced, especially for unseen horizons.

These results emphasize the importance of customizing RoPE's period coefficients settings and utilizing tunable RoPE to enable flexible and accurate forecasting in time series analysis. Appendix E offers visualizations demonstrating how different initial ranges impact the frequency components, along with a detailed analysis of the distribution of RoPE periods optimized for each dataset.

### D.5 The Impact of Patch Size Selection

These experiments highlight the critical impact of patch size selection, particularly in varied-horizon forecasting scenarios. As demonstrated in Figure 10, various combinations of training and forecasting horizons exhibit distinct preferences for patch sizes. For instance, when the training forecasting horizon is 720, during the inference stage, longer forecasting horizons prefer larger patch sizes. Conversely, on shorter training horizons, such as 96 and 192, choosing large patch sizes for longer horizons can lead to performance collapse. This difference underscores the complexity and necessity of optimal patch size selection in achieving effective elastic forecasting.

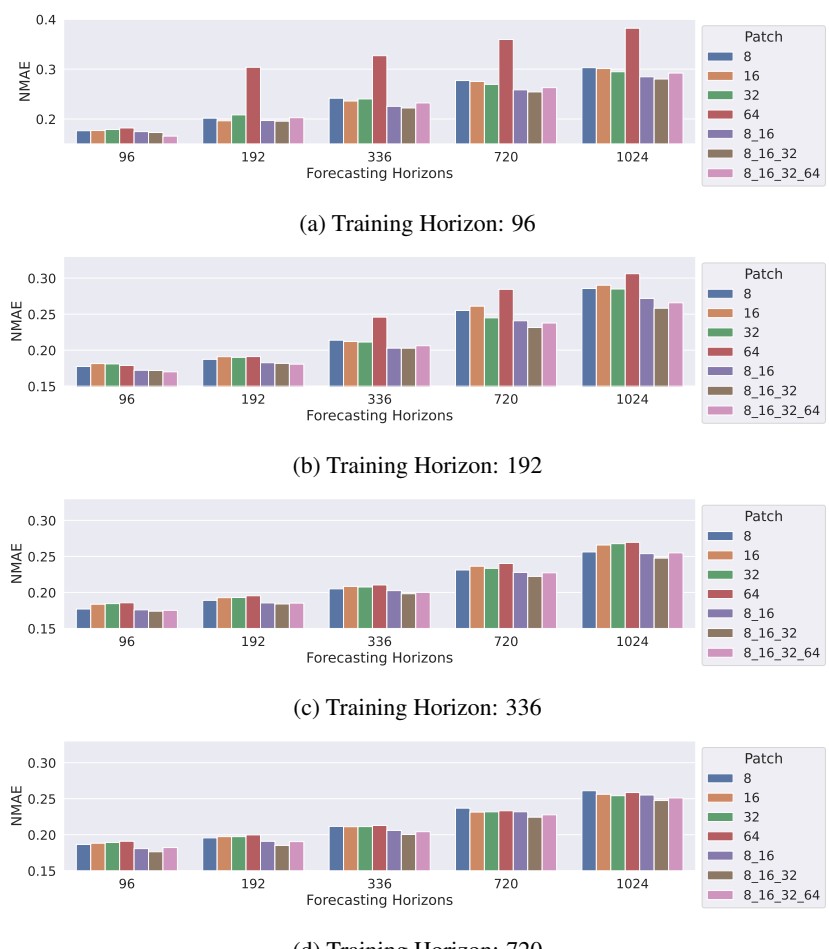

(a) Training Horizon: 96

(b) Training Horizon: 192

(c) Training Horizon: 336

(d) Training Horizon: 720

Figure 10: The performance of difference patch size selection.

Moreover, Figure 10 clearly show that multi-patch configurations typically surpass single patch sizes across most forecasting horizons. The configuration $\boldsymbol{p} = \{8, 16, 32\}$ consistently offers the lowest NMAE values, striking an optimal balance between capturing short-term dynamics and long-term trends. However, introducing larger patches ($\boldsymbol{p} = \{8, 16, 32, 64\}$) does not always enhance performance and can sometimes increase the NMAE, indicating that overly complex configurations may not yield additional benefits and could be counterproductive.

These findings emphasize the advantages of employing multi-patch configurations to improve the accuracy of varied-horizon forecasting. They also highlights the importance of carefully selecting patch size combinations to optimize performance and minimize computational expenses.

## E  Visualization of Periods in RoPE

### E.1  Initialized Periods

To provide a more intuitive visualization of the initial distribution of periodicity coefficients, we present this in Figure 11.

### E.2  Tuned periodicity coefficients

In Figure 12, we present the distribution of tuned periodicity coefficients for each dataset.

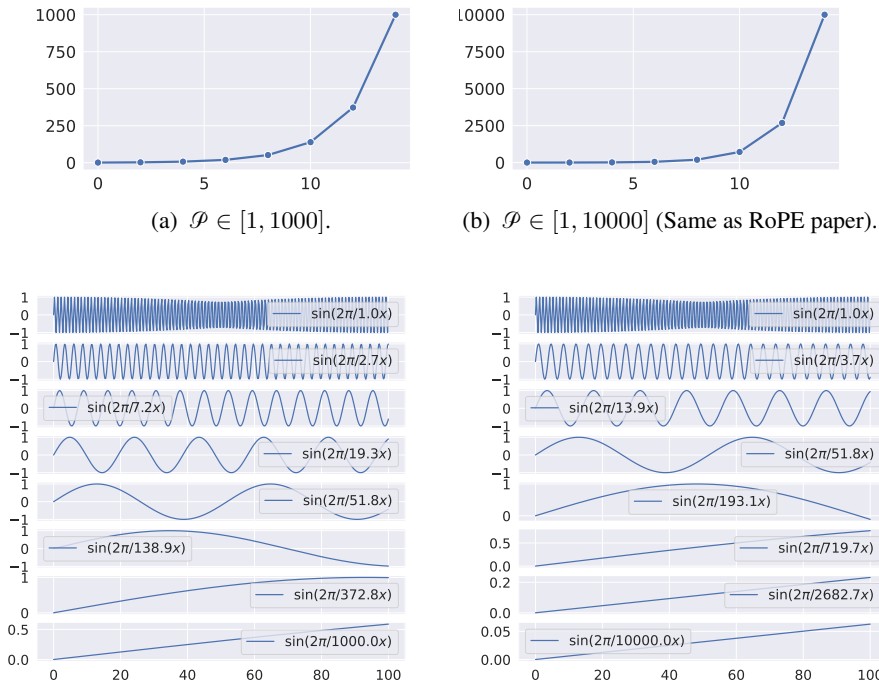

(a) $\mathcal{P} \in [1, 1000]$.  (b) $\mathcal{P} \in [1, 10000]$ (Same as RoPE paper).

(c) Frequency distribution when $\mathcal{P} \in [1, 1000]$.  (d) Frequency distribution when $\mathcal{P} \in [1, 10000]$ (Same as RoPE paper).

Figure 11: The initialized periodicity coefficients in RoPE. Here we set the dimension $d$ to 16.

# F  Model Efficiency

## F.1  Overall Computational Efficiency Comparison

Table 6: Computation memory. The batch size is 1 and the prediction horizon is set to 96.

| Metric | Dataset | DLinear | PatchTST | Autoformer | iTransformer | ElasTST |
|---|---|---|---|---|---|---|
| NPARAMS (MB) | ETTm1 | 0.075 | 2.145 | 23.273 | 3.366 | 2.240 |
| | Electricity | 0.076 | 2.146 | 29.701 | 3.366 | 2.240 |
| | Traffic | 0.078 | 2.149 | 40.783 | 3.366 | 2.240 |
| | Weather | 0.075 | 2.145 | 23.560 | 3.366 | 2.240 |
| | Exchange | 0.075 | 0.135 | 23.286 | 3.366 | 2.240 |
| Max GPU Mem. (GB) | ETTm1 | 0.002 | 0.009 | 0.227 | 0.026 | 0.024 |
| | Electricity | 0.060 | 0.068 | 0.246 | 0.037 | 0.112 |
| | Traffic | 0.161 | 0.168 | 0.279 | 0.157 | 0.263 |
| | Weather | 0.004 | 0.012 | 0.228 | 0.027 | 0.028 |
| | Exchange | 0.002 | 0.002 | 0.227 | 0.026 | 0.024 |

In Table 6, we compare the memory usage of ElasTST with other baseline models. The comparison reveals that ElasTST does not require more computational resources than other Transformer-based models.

## F.2  Memory Usage Introduced by TRoPE

In Table 7, we compare the memory usage of ElasTST with different position encodings. The results show that RoPE adds an almost negligible number of parameters compared to vanilla absolute position encoding. While applying rotation matrix does introduce slightly higher memory usage, it remains within acceptable limits.

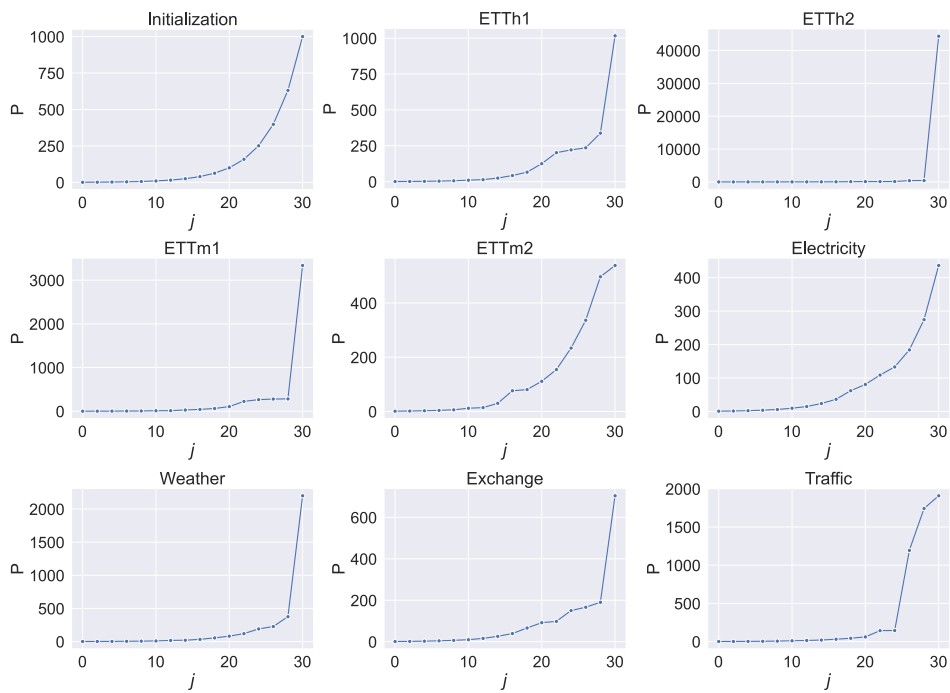

Figure 12: Tuned periodicity coefficients over different datasets.

Table 7: Memory consumption of ElasTST using different position encoding approaches. The batch size is 1 and the forecasting horizon is 1024.

| Metric | Abs PE | RoPE w/o Tunable | Tunable RoPE |
|---|---|---|---|
| Max GPU Mem. (GB) | 0.0480 | 0.0539 | 0.0539 |
| NPARAMS (MB) | 5.1225 | 5.1225 | 5.1228 |

### F.3 Memory Usage Introduced by Multi-scale Patch Assembly

Table 8: Memory consumption under different patch size settings. The batch size is 1 and the forecasting horizon is 1024.

| Metric | p=1 | p=8 | p=16 | p=32 | p={1,8,16,32} | p={8,16,32} |
|---|---|---|---|---|---|---|
| Max GPU Mem. (GB) | 0.6747 | 0.0536 | 0.0415 | 0.0360 | 0.6751 | 0.0539 |
| NPARAMS (MB) | 5.0130 | 5.0267 | 5.0424 | 5.0738 | 5.1257 | 5.1228 |

While our model uses a shared Transformer backbone to process all patch sizes, this can be done either in parallel or sequentially, depending on whether shorter computation times or lower memory usage is prioritized. In practice, we chose the sequential approach, where each patch size is processed individually, and the total forecast is assembled afterward. This approach ensures that the memory bottleneck depends on the smallest patch size used.

As indicated in Table 8, memory usage is primarily influenced by the smallest patch size, not by the number of patch sizes employed. The additional parameters introduced by using multiple patch sizes are almost negligible. For example, the multi-patch setting 8,16,32 used in the paper requires the same maximum memory as using a single patch size of 8. Under resource constraints, the minimum patch size can be adjusted to balance model performance and memory usage.

