# OpenReview forum: "ElasTST: Towards Robust Varied-Horizon Forecasting with Elastic Time-Series Transformer"
_NeurIPS.cc/2024/Conference — NeurIPS 2024 poster_

### Official Review · Reviewer_i9JA · 2024-07-05

**Soundness:** 2
**Presentation:** 2
**Contribution:** 2
**Rating:** 5
**Confidence:** 5

**Summary:**

The paper introduces a masked encoder based transformer model for time series forecasting. It performs masking in the observation space, replacing the prediction horizon with 0s. It further applies an attention mask such that tokens only attend to the context, and do not attend to the masked tokens. The paper also uses a tunable version of rotary positional embeddings, and ensembles predictions over various patch sizes.

**Strengths:**

The paper proposes a method which can adapt to multiple forecast horizons, which is a weakness of many recent deep forecasting models. The paper presents several empirical analyses to verify the claims made, and also presents a detailed look at related work.

**Weaknesses:**

Overall, the proposed approach is very similar to Moirai, with several minor changes:

i) Perform the masking operation in observation space

ii) Add an attention mask

iii) Tunable rotary embeddings

iv) Ensembling results over multiple patch sizes

However, it seems that even more features which Moirai had were removed, making it a weaker proposition. Such features include probabilistic predictions, multivariate capabilities, and any context length. Furthermore, few experiments are done to show the benefits of the changes, such as (i), (ii), and (iv).

Others:
1. The statement, "we find that on the challenging datasets such as Weather and Electricity, dataset-specific tuning still offers advantages" (L238-L239) is a little eyebrow raising, in the sense that it is stating the obvious. I believe most researchers expect in-distribution predictions to outperform zero-shot predictions. Such statements should be rephrased.
2. Effects of tuning rotary embeddings in figure 4 seem very marginal, error margins should be provided for such cases.

**Questions:**

None

**Limitations:**

Authors have provided limitations of their work.

---

> ### Author Rebuttal · Authors · 2024-08-06
>
> Thank you for your in-depth reviews and constructive suggestions. We appreciate the opportunity to clarify the distinctions between our approach and MOIRAI, as well as highlight the key contributions of our paper.
>
> While there are similarities between our model and MOIRAI due to shared influences from image/video generation Transformer frameworks such as DiT [1] and SORA [2], there are significant differences in our objectives and approaches.
>
> MOIRAI focuses on **universal forecasting, emphasizing pre-training and transferability across datasets**. It introduces any-variate attention and probabilistic predictions to enhance these capabilities. However, these features do not directly address the challenge of robust forecasting across arbitrary horizons. Moreover, while MOIRAI incorporates techniques from recent LLM architectures, such as vanilla RoPE, RMSNorm, and SwiGLU, these were applied without specific adaptations for time series. As shown in the global response (Figure 3), using vanilla RoPE directly in varied-horizon forecasting still presents significant challenges.
>
> Our work **specifically targets robust varied-horizon forecasting** with three key designs:
>
> (i) structured self-attention masks for consistent outputs across different forecasting horizons,
>
> (ii) tunable rotary position embeddings for adapting to varying periodic patterns, and
>
> (iii) multi-scale patch assembly to balance patch settings preferred by different horizons.
>
> The full ablation study in the global response clearly demonstrates that **each of these designs plays a crucial role in enabling robust varied-horizon forecasting**:
>
> (i) Structured attention masks ensure reliable inference on horizons that differ from the training horizon (Figures 1 and 2).
>
> (ii) Tunable RoPE is crucial for enhancing the model’s extrapolation capability (Figure 3).
>
> (iii) Multi-scale patch assembly reduces sensitivity to patch size hyper-parameters across different horizons (Figure 4).
>
> While our model also supports any context length, this feature was not highlighted in the manuscript as it is less central to our key technical contributions.
>
> In summary, ElasTST and MOIRAI address different challenges in time series forecasting, each offering solutions to distinct problems. In the future, our methods could potentially be integrated within a broader framework to enhance both generality and robustness in time series forecasting.
>
> ----------
> **Improve the presentation of this paper**
>
> > 1. The statement, "we find that on the challenging datasets such as Weather and Electricity, dataset-specific tuning still offers advantages" (L238-L239) is a little eyebrow raising, in the sense that it is stating the obvious. I believe most researchers expect in-distribution predictions to outperform zero-shot predictions. Such statements should be rephrased.
> > 2. Effects of tuning rotary embeddings in figure 4 seem very marginal, error margins should be provided for such cases.
>
> Thank you for your valuable suggestions.
>
> We agree that some statements could be better phrased. Specifically, regarding the statement in L238-L239, our intention was to emphasize that while the zero-shot capability of foundation models is impressive, additional dataset-specific tuning may still be necessary for challenging datasets to achieve optimal forecasting performance. We will revise these statements in the manuscript to convey our findings more clearly and accurately.
>
> In the experimental section, we will incorporate key results from the global response into the main text and add error margins to Figure 4 to improve result representation.
>
> We appreciate your feedback and will refine the paper accordingly to ensure a clearer and more thorough presentation in the revised manuscript.
>
> [1] Peebles, W., & Xie, S. (2023). Scalable diffusion models with transformers. *ICCV.*
>
> [2] Tim Brooks et. al., (2024). Video generation models as world simulators.

---

> ### Comment · Area_Chair_nbPY · 2024-08-12
> **author reviewer discussion**
>
> Dear reviewer,
>
> The author-reviewer discussion ends soon. If you need additional clarifications from the authors, please respond to the authors asap. Thank you very much.
>
> Best,
>
> AC

---

> > ### Comment · Reviewer_i9JA · 2024-08-12
> >
> > Thank you authors for the rebuttal. I understand that this paper focuses on the varied horizon setting. Moirai has addressed this with a simple approach - during training, train with multiple horizon lengths. This seems to be a simple method which should be compared to. Also, I do not see evidence in the paper indicating that the structured mask is critical for varied horizon prediction. As such, my rating remains.

---

> > > ### Author Response · Authors · 2024-08-13
> > > **Response to Official Comment by Reviewer i9JA**
> > >
> > > Dear Reviewer i9JA,
> > >
> > > Thank you for your response, and we appreciate the opportunity to address your concerns. We believe that our previous responses have covered these issues.
> > >
> > > > Moirai has addressed this with a simple approach - during training, train with multiple horizon lengths. This seems to be a simple method which should be compared to.
> > > >
> > >
> > > MOIRAI’s training approach (with maximum forecasting horizon of 256) does not guarantee robust extrapolation capabilities, while the design in ElasTST is specifically aimed at improving performance on unseen horizons.
> > >
> > > As shown in Figures 1 and 3 in the global response PDF, stable performance may be achieved within the training range (1-336) even without certain design elements, but these elements become crucial when dealing with longer, unseen horizons. Simply training with multiple horizon lengths alone does not ensure robustness over extended inference horizons.
> > >
> > > > Also, I do not see evidence in the paper indicating that the structured mask is critical for varied horizon prediction.
> > > >
> > >
> > > The structured self-attention masks ensure consistent outputs when inference horizons differ from the training horizon. This is evidenced in the global response PDF.
> > >
> > > Specifically, Figure 1 shows that removing the structured attention mask (w/o Mask) can lead to issues when forecasting horizons outside the training range, particularly noticeable in the weather dataset. Figure 2 further illustrates that the absence of the structured mask leads to a performance drop for both shorter and longer inference horizons compared to the full model (ElasTST).
> > >
> > > We will incorporate these experimental results and analyses into future revisions to clarify our contributions. We hope our response has addressed your concerns, and we welcome any further questions or suggestion you may have.
> > >
> > > Best Regards,
> > >
> > > All Authors

---

### Official Review · Reviewer_eA7o · 2024-07-12

**Soundness:** 2
**Presentation:** 2
**Contribution:** 2
**Rating:** 5
**Confidence:** 3

**Summary:**

This paper introduces ElasTST, which aimed at addressing the challenge of varied-horizon time-series forecasting. ElasTST integrates a non-autoregressive architecture with placeholders for forecasting horizons, structured self-attention masks, and a tunable rotary position embedding. Furthermore, it employs a multi-scale patch design to incorporate both fine-grained and coarse-grained temporal information, enhancing the model’s adaptability across different forecasting horizons. Experiments in the paper demonstrate the model's performance in providing robust predictions when extrapolating beyond the trained horizons.

**Strengths:**

- The paper attempts to address an important issue in time series applications by enabling the model to adapt to varied forecasting horizons during the inference stage, which can make the trained forecasting model more flexible in usage.

- Introducing multi-scale patching operations to construct input tokens for the Transformer can effectively handle local patterns of different granularities.

- The paper introduces a tunable Rotary Position Embedding (RoPE) module tailored for time series tasks, showing potential for broader application across various forecasting methods.

**Weaknesses:**

- The newly proposed tunable RoPE module lacks comparative analysis with previous methods. A comparison with methods such as no positional embedding, vanilla positional encoding, trainable positional embedding matrices, and original rotary positional embeddings, would clarify the benefits of different approaches. Since tunable RoPE appears easily transferable, demonstrating its performance enhancement by applying it to previous models like PatchTST and Autoformer would provide more intuition.

- Using longer sequence length significantly increases the computational cost in Transformers. Comparing with baselines like PatchTST, the paper uses multiple patch sizes as input processing method, which significantly increases computational costs. Further analysis like how much adding a patch size can increase computational costs or what patch size combination are more reasonable to choose under resource constraints would be beneficial.

**Questions:**

- Given the significant increase in computational cost from longer token sequences, would independently inputting each patch size into the transformer layers, instead of all at once, reduce memory usage while maintaining model performance?

- With multiple patch sizes used in the study, how are the RoPE applied? Are they applied independently to tokens from each patch size, or collectively to all tokens from different patch sizes?

- Ablation studies indicate that incorporating smaller patch sizes seems to benefit model performance substantially. If my understanding is correct, when the patch size is 1 (excluding tunable RoPE), the structure is similar to the decoder part of models like Informer and Autoformer. Would using a patch size of 1 worsen the model performance? Would adding "1_" to the existing best patch size setting "8_16_32" improve performance? Using patch size 1 turns the model into a channel-independent Informer-decoder-like model, which significantly increases computational costs, hence it you don't have to consider this on large datasets.

---

> ### Author Rebuttal · Authors · 2024-08-06
>
> Thanks for your in-depth reviews, insightful questions, and constructive suggestions. We hope the following responses can help to address all your questions and concerns.
>
> ## Response to Weakness 1
>
> > A comparison with methods such as no positional embedding, vanilla positional encoding, trainable positional embedding matrices, and original rotary positional embeddings, would clarify the benefits of different approaches. Since tunable RoPE appears easily transferable, demonstrating its performance enhancement by applying it to previous models like PatchTST and Autoformer would provide more intuition.
> >
>
> Thank you for highlighting this gap in our experiments. We have updated Figure 3 in the global response to include the performance of ElasTST with various positional embeddings. The results demonstrate that Tunable RoPE significantly enhances the model’s extrapolation capabilities. While other positional embeddings perform well on seen horizons, they struggle when the forecasting horizon extends beyond the training range. Although the original RoPE performs well in NLP tasks, it falls short in time series forecasting. Dynamically tuning RoPE parameters based on the dataset’s periodic patterns proves highly beneficial, especially for unseen horizons, with the benefits becoming more pronounced as the inference horizon lengthens.
>
> We appreciate your suggestion to apply Tunable RoPE to previous models like PatchTST and Autoformer. However, PatchTST requires per-horizon training and deployment, making it difficult to showcase the benefits of Tunable RoPE since it cannot handle longer inference horizons once trained for a specific horizon. Essentially, our architecture—after removing the tunable RoPE and multi-patch designs—can be viewed as a version of PatchTST that adapts to varied horizons. As for Autoformer, it replaces the self-attention mechanism with an auto-correlation mechanism that inherently captures sequential information, making positional embeddings unnecessary.
>
> Your feedback has been invaluable in refining our work, and we will update the manuscript to include these discussions.
>
> ## Response to Weakness 2
>
> > Further analysis like how much adding a patch size can increase computational costs or what patch size combination are more reasonable to choose under resource constraints would be beneficial.
> >
>
> ElasTST uses a shared Transformer backbone to process all patch sizes, which can be done either in parallel or sequentially, depending on whether shorter computation times or lower memory usage are prioritized. We chose sequential processing, where each patch size is handled individually, keeping the memory consumption comparable to baselines like PatchTST (see Table 1 in global response).
>
> In this approach, memory usage is primarily influenced by the smallest patch size, not the number of patch sizes used (see Table 3 in global response). For example, the multi-patch setting in the paper (sizes {8,16,32}) requires the same maximum memory as using a single patch size of 8. Under resource constraints, adjusting the minimum patch size helps balance model performance and memory usage. We provide further analysis on the performance of different patch size combinations in the response to Question 3.
>
> ## Response to Question 1
>
> > Given the significant increase in computational cost from longer token sequences, would independently inputting each patch size into the transformer layers, instead of all at once, reduce memory usage while maintaining model performance?
> >
>
> Yes, as mentioned in our response to Weakness 2, independently inputting each patch size into the transformer layers can indeed reduce memory usage without degrading model performance.
>
> ## Response to Question 2
>
> > With multiple patch sizes used in the study, how are the RoPE applied? Are they applied independently to tokens from each patch size, or collectively to all tokens from different patch sizes?
> >
>
> RoPE is applied independently to each patch size, and tokens from different patch sizes do not interact within the Transformer layer.
>
> ## Response to Question 3
>
> > Ablation studies indicate that incorporating smaller patch sizes seems to benefit model performance substantially. Would using a patch size of 1 worsen the model performance? Would adding "1_" to the existing best patch size setting "8_16_32" improve performance?
> >
>
> Thank you for your questions. We would like to clarify that smaller patch sizes do not always improve model performance. In Appendix C.3 of the submitted manuscript, we discuss how different patch size combinations impact performance across various training horizons. For instance, with a training horizon of 720, a smaller patch size can sometimes result in poorer performance.
>
> In Figure 4 of the global response, we included the case with a patch size of 1 for a more comprehensive comparison. The results indicate that using a patch size of 1, or adding it to a multi-patch combination, does not consistently lead to better or worse performance. The optimal patch size varies by dataset and forecasting horizon. Overall, the 8_16_32 combination provides a good balance between performance and memory consumption for varied-horizon forecasting.
>
> We sincerely appreciate your suggestions and feedback. We will revise the paper to clarify the workings of Tunable RoPE and multi-patch assembly, and we will include a more comprehensive comparative and computational analysis to provide a holistic view of the proposed model.

---

> > ### Comment · Reviewer_eA7o · 2024-08-14
> >
> > Thanks to the authors for the rebuttal. I will keep my positive rating.

---

### Official Review · Reviewer_qEB8 · 2024-07-13

**Soundness:** 2
**Presentation:** 3
**Contribution:** 2
**Rating:** 5
**Confidence:** 5

**Summary:**

The paper introduces ElasTST, a novel approach designed to enhance robustness in forecasting across various horizons. The proposed architecture dynamically adapts to different forecasting horizons, addressing a significant challenge in the literature. Previous methods typically relied on recursive approaches or utilized masks that failed to maintain horizon invariability. ElasTST aims to achieve high forecasting accuracy while ensuring adaptability across different horizons by employing a transformer encoder as its backbone. The approach incorporates structured attention masks through patching and rotary position embeddings (RoPE) to encode relative positional information for improved forecasting.

**Strengths:**

1. The paper effectively tackles a crucial issue in the forecasting literature—enhancing deep learning architectures to produce forecasts for any horizon while maintaining horizon invariability and reducing the concatenation of errors.
2. The approach leverages well-established research, such as patching from the PatchTST paper, and also rotary position embeddings, to handle long-horizon forecasting tasks.
3. The paper considers recent advancements in foundation models for time series forecasting.

**Weaknesses:**

1. The literature review on foundational models is lacking. Notable models such as TimeGPT-1 [1], Chronos [2], TinyTimeMixers [3], and Moment [4] are not discussed.
2. A traditional encoder transformer is not included as a baseline, which would help demonstrate the improvements offered by the proposed method. Additionally, NLP-based architectures such as NHITS [5] and TsMixer [6] are absent from the comparisons.
3. The paper does not adequately address the computational scalability of ElasTST when applied to very large datasets or extremely long time series.
4. The authors modified the traditional metrics used in long-horizon literature (MSE and MAE) to scaled metrics. They argue this adjustment reports fairer results, but it overlooks the fact that these datasets are typically already scaled, making additional scaling in the metric unnecessary.
5. The paper lacks a detailed discussion on potential failure cases or scenarios where ElasTST might underperform, which would provide a more balanced view of its applicability.

[1] https://arxiv.org/abs/2310.03589
[2] https://arxiv.org/abs/2403.07815v1
[3] https://arxiv.org/abs/2401.03955
[4] https://arxiv.org/abs/2402.03885
[5] https://arxiv.org/abs/2201.12886
[6] https://arxiv.org/abs/2303.06053

**Questions:**

1. What is the computational complexity added by the Rotary Position Embeddings (RoPE)?
2. In some cases, the results are very similar to PatchTST, such as with the ETTm2 dataset. Is there an intuition or explanation regarding the types of data that explain these similarities and differences in performance?
3. It would be interesting to compare the performance gains for each point in the forecasting window. Are the gains through the whole window or in the earliest/latest points?

**Limitations:**

Yes

---

> ### Author Rebuttal · Authors · 2024-08-06
>
> Thanks for your in-depth reviews, insightful questions, and constructive suggestions. We hope the following responses can help to address all your questions and concerns.
>
> ## Response to Weakness 1
>
> We will expand our literature review to provide a more comprehensive discussion of recent time series foundational models. We briefly summarize these models in the table below. Most of these efforts employ standard architecture designs, position encodings, and patching approaches. However, they often lack in-depth investigation into the challenges of generating robust forecasts across varied horizons. Our work directly addresses this gap by enhancing model design for varied-horizon robustness. Detailed discussions will be incorporated into the revision.
>
> | Model | Backbone | Dec. Scheme. | Pos. Emb. | Token. |
> | --- | --- | --- | --- | --- |
> | TimeGPT-1 | Enc-Dec Transformer | AR | Abs PE | - |
> | Chronos | Enc-Dec Transformer | AR | Simplified relative PE | Quantization |
> | TimesFM | Decoder-only Transformer | AR | Abs PE | Patching |
> | DAM | Transformer Encoder | NAR | Abs PE | ToME |
> | Tiny Time Mixers | TSMixer | NAR | - | Patching |
> | MOIRAI | Transformer Encoder | NAR | RoPE | Patching |
> | MOMENT | Transformer Encoder | NAR | Learnable relative PE | Patching |
>
> ## Response to Weakness 2
>
> We acknowledge the importance of including a more comprehensive baseline. Due to the word limitation, we have provided the averaged performance of these baselines across all datasets in the table below for reference. The revised manuscript will include detailed results and analysis to provide a more thorough evaluation.
>
> | Infer Hor. | ElasTST (NMAE) | NHiTS (NMAE) | TSMixer (NMAE) | TransformerEnc (NMAE) |
> | --- | --- | --- | --- | --- |
> | 96 | 0.16775 | 0.205125 | 0.25375 | 0.2055 |
> | 192 | 0.177875 | 0.220375 | 0.2645 | 0.219375 |
> | 336 | 0.188125 | 0.247875 | 0.283 | 0.23375 |
> | 720 | 0.21375 | 0.29075 | 0.30675 | 0.2505 |
>
> ## Response to Weakness 3
>
> Thank you for raising this important point. ElasTST introduces minimal additional computation compared to other non-autoregressive Transformer-based models, as shown in Table 1 in the global response. However, we acknowledge the need for further scalability improvements. To address this, we are exploring solutions such as dimensionality reduction, quantization, and memory-efficient attention mechanisms to reduce computational overhead for large datasets.
>
> ## Response to Weakness 4
>
> Thank you for your observation regarding the evaluation metrics. As noted in Appendix B.3, some existing models apply z-score standardization during data preprocessing and compute metrics before reversing the standardization, which can complicate direct comparisons when different scaler are used. We chose NMAE (Normalized Deviation in the M4 and M5 literature) as our primary evaluation metric because it is scale-insensitive and widely accepted in recent studies [1].
>
> Additionally, we have also calculated MAE and MSE both before and after de-standardization. These results will be included in the Appendix in future revisions.
>
> [1] Oreshkin, B. N., Carpov, D., Chapados, N., & Bengio, Y. (2020). N-BEATS: Neural basis expansion analysis for interpretable time series forecasting. *ICLR*.
>
> ## Response to Weakness 5
>
> While our method improves varied-horizon forecasting, as shown in Figure 3, the training horizon is a tunable hyper-parameter, and the optimal value varies across datasets. An inappropriate choice, like 96, can significantly degrade performance. Besides, the selected horizon of 336 used in the main results is not always optimal, suggesting that tailoring the training horizon for each dataset can further enhance the model’s effectiveness.
>
> ## Response to Question 1
>
> In Table 2 of the global response, we compare the memory usage of ElasTST with different positional encodings. The results show that RoPE adds a negligible number of parameters compared to vanilla absolute position encoding. Although applying the rotation matrix introduces slightly higher memory usage, it remains acceptable.
>
> ## Response to Question 2
>
> Thank you for your insightful observation. To understand the performance similarities, we analyzed the trend and seasonality of these datasets [1]. We find that the similarities between ElasTST and PatchTST could be attributed to varying levels of seasonality. Specifically, ETTm1 and ETTm2 have minor seasonality, accompanying similar performance, while ETTh1 and ETTh2 exhibit stronger seasonal patterns, where ElasTST shows clear advantages.
>
> Without the additional designs for robust varied-horizon forecasting, ElasTST essentially functions as a variable-length version of PatchTST. Therefore, the differences in performance could reflect the datasets’ sensitivity to our specific architecture designs.
>
> |  | ETTh1 | ETTh2 | ETTm1 | ETTm2 |
> | --- | --- | --- | --- | --- |
> | Seasonality | 0.4772 | 0.3608 | 0.0105 | 0.0612 |
>
> [1] Wang, X., Smith, K. A., & Hyndman, R. J. (2006). Characteristic-based clustering for time series data. *Data Mining and Knowledge Discovery*, *13*(3), 335–364.
>
> ## Response to Question 3
>
> In Figure 2 of the global response, we compare the performance gains of each model design across different points within the forecasting window. The benefits of structured attention masks are consistent throughout the entire horizon, while the advantages of tunable RoPE and multi-scale patch assembly become more pronounced when dealing with unseen horizons. Notably, tunable RoPE significantly enhances the model’s extrapolation capability. Due to the word limitation, we will include a more detailed analysis in the revised version.

---

### Author Rebuttal · Authors · 2024-08-06

We sincerely thank the reviewers for their insightful comments.

In response to your suggestions, we have conducted additional experiments, which are detailed in the attached PDF. The ablation study further confirms that each proposed design element is crucial for achieving robust varied-horizon forecasting.

Specifically, the supplementary experiments include:

1. The benefits of each proposed design (Figure 1).
2. Performance gains across the forecasting horizon for each model design (Figure 2).
3. A detailed comparison of tunable RoPE with other positional embedding methods (Figure 3).
4. The performance of different multi-patch size combinations, including the case with a patch size of 1 (Figure 4).

To address concerns about computational efficiency, we have provided analysis in Tables 1, 2, and 3, where we compare our model to other baselines and calculate the additional memory consumption introduced by tunable RoPE and multi-scale patch assembly. Our analysis shows that ElasTST adds minimal computational overhead compared to other non-autoregressive Transformer models, and the inclusion of tunable RoPE and multi-patch design has a negligible impact on overall efficiency.

We will include these experiments and analyses in the revised paper, and we welcome any further questions or suggestions.

Table 1. Memory consumption of ElasTST and baselines. The batch size is 1 and the horizon is set to 1024.

|  | TransformerEnc | Autoformer | PatchTST | ElasTST |
| --- | --- | --- | --- | --- |
| Max GPU Mem. (GB) | 0.1516 | 0.1678 | 0.0389 | 0.0539 |
| NPARAMS (MB) | 3.4151 | 5.5977 | 7.0337 | 5.1228 |

Table 2. Memory consumption of ElasTST that using different position encoding approaches. The batch size is 1 and the forecasting horizon is 1024.

|  | Abs PE | RoPE w/o Tunable | Tunable RoPE  |
| --- | --- | --- | --- |
| Max GPU Mem. (GB) | 0.048 | 0.0539 | 0.0539 |
| NPARAMS (MB) | 5.1225 | 5.1225 | 5.1228 |

Table 3. Memory consumption under different patch size settings. The batch size is 1 and the forecasting horizon is 1024.

|  | p=1 | p=8 | p=16 | p=32 | p={1,8,16,32} | p={8,16,32} |
| --- | --- | --- | --- | --- | --- | --- |
| Max GPU Mem. (GB) | 0.6747 | 0.0536 | 0.0415 | 0.036 | 0.6751 | 0.0539 |
| NPARAMS (MB) | 5.013 | 5.0267 | 5.0424 | 5.0738 | 5.1257 | 5.1228 |

---

### Decision · Program_Chairs · 2024-09-25

**Decision:**

Accept (poster)

**Comment:**

This paper proposes a non-autoregressive architecture for varied-horizon forecasting.  The approach incorporates structured attention masks through patching and rotary position embeddings (RoPE) to encode relative positional information for improved forecasting. This paper addresses a critical issue in time series forecasting and the proposed method is novel and effective. There are concerns about the scalability of the proposed method, but the authors have provided reasonable responses to address these concerns. Overall, this is a solid paper and I recommend it for acceptance.